# Life history predicts global population responses to the weather in terrestrial mammals

**John Jackson[1,2]\*, Christie Le Coeur[3], Owen Jones[1]**

[1]Interdisciplinary centre for population dynamics (CPop), Department of Biology, University of Southern Denmark, Odense, Denmark; [2]Department of Zoology, University of Oxford, Oxford, United Kingdom; [3]Centre for Ecological and Evolutionary Synthesis (CEES), Department of Biosciences, University of Oslo, Oslo, Norway

**Abstract** With the looming threat of abrupt ecological disruption due to a changing climate, predicting which species are most vulnerable to environmental change is critical. The life-history of a species is an evolved response to its environmental context, and therefore a promising candidate for explaining differences in climate-change responses. However, we need broad empirical assessments from across the world's ecosystems to explore the link between life history and climate-change responses. Here, we use long-term abundance records from 157 species of terrestrial mammals and a two-step Bayesian meta-regression framework to investigate the link between annual weather anomalies, population growth rates, and species-level life history. Overall, we found no directional effect of temperature or precipitation anomalies or variance on annual population growth rates. Furthermore, population responses to weather anomalies were not predicted by phylogenetic covariance, and instead there was more variability in weather responses for populations within a species. Crucially, however, long-lived mammals with smaller litter sizes had smaller absolute population responses to weather anomalies compared with their shorter living counterparts with larger litters. These results highlight the role of species-level life history in driving responses to the environment.

## Editor's evaluation

Using 486 long-term population records of 157 mammal species, the authors show that species with a short life span and large litters are more affected, either positively or negatively, by extreme weather events than are species with a long life span and few offspring. This suggests that these "fast" species may require particular conservation attention, to avoid extinction due to the increased frequency and magnitude of extreme events.

## Introduction

Climate change is one of the greatest challenges we face in the twenty-first century (*Diaz, 2019*). Although habitat loss and direct exploitation are currently the most important drivers of extinction in the natural world (*Daskalova et al., 2020a*; *Diaz, 2019*), changes to the climate altering both the mean and variance of weather conditions and the frequency of extreme events, are predicted to cause widespread declines of global biodiversity in the coming decades (*Almond et al., 2020*; *Newbold, 2018*; *Soroye et al., 2020*; *Thomas et al., 2004*). For mammals and birds, temperature increases are already associated with declining population trends (*Spooner et al., 2018*) and many endangered species have already been negatively impacted by climate change in at least part of their range

**\*For correspondence:**
john.jackson@zoo.ox.ac.uk

**Competing interest:** The authors declare that no competing interests exist.

(*Pacifici et al., 2017*; *Pacifici et al., 2020*). Perhaps more worryingly, abrupt ecological disruption due to climate change has been predicted to have large negative future impacts on biodiversity, with tropical ecosystems being affected as early as 2030 (*Newbold, 2018*; *Trisos et al., 2020*). Furthermore, these future impacts will likely be exacerbated by interactions between the climate and other drivers of extinction such as habitat loss or disease prevalence (*Brook et al., 2008*; *Cohen et al., 2020*; *Williams et al., 2019*). Research highlighting the species and ecosystems that are most sensitive to climate-change impacts will therefore provide crucial knowledge to prevent future losses to global biodiversity.

Not all species are equally sensitive to changes in the climate. Species vary in their climatic niches and in their behavioural, physiological, and demographic responses to environmental change and we therefore expect there to be both climate 'winners' and 'losers' (*Antão et al., 2020*; *Bellard et al., 2012*; *Moritz and Agudo, 2013*). At the macro scale, species-occupancy data highlight that geographic range shifts are the key response associated with climate change across taxa, resulting in changes to community composition, but not necessarily population decline (*Antão et al., 2020*; *Chen et al., 2011*; *Dornelas et al., 2019*). Furthermore, this pattern also occurs in broad species-assemblage and abundance-change data (*Dornelas et al., 2019*; *Leung et al., 2020*). While there is a lack of consistent directional temperature-related biodiversity change effects today (*Antão et al., 2020*), many species are predicted to become at risk of climate-induced population declines before 2,100 (*Trisos et al., 2020*). Therefore, investigating the mechanisms of how climate change induces population change may improve our predictions of the species most-affected by warming temperatures.

Understanding the mechanisms underpinning population declines in response to climate change and the temporal scale at which these mechanisms occur, at short- or long-term timescales, remains one of the main challenges when investing the influence of climate change on population change. Population abundance time-series and demographic data have begun to reveal how climate change leads to population decline. For birds and mammals average abundance trends were negatively associated with long-term rates of climate warming (*Spooner et al., 2018*), but sparse data in short time-series are at risk of overfitting weather effects (*Knape and de Valpine, 2011*). Therefore, targeted demographic studies unpicking how changes in weather patterns cause population change (*Cordes et al., 2020*; *Coulson et al., 2001*; *Layton-Matthews et al., 2021*; *Paniw et al., 2019*; *Paniw et al., 2021*; *Woodroffe et al., 2017*) and whether species traits can help predict these changes (*Pacifici et al., 2017*) are vital to highlight vulnerable species. While emphasis is often on long-term temperature trends, the immediate impact of the weather on populations, particularly in the context of extreme or anomalous events (e.g. heat waves and droughts), is also important (*Maxwell et al., 2018*). Furthermore, there may also be impacts of increased variance in weather conditions as opposed to changes in central tendency (*Lawson et al., 2015*; *Le Coeur et al., 2021*; *Stenseth et al., 2003*). Applying these concepts at a comparative scale and assessing finer-scale population changes with respect to changes in the weather (and particularly extreme weather events or weather variance), and their relationship to species traits, will aid in illuminating climate-change responses across the tree of life (*Compagnoni et al., 2021*; *Paniw et al., 2021*).

Life-history variation is a promising factor that could explain observed variation in responses to climate change (*Pacifici et al., 2017*). The timing of key demographic events of survival and recruitment across the life cycle, or life-history traits, are evolved responses to the environment, and characteristics relating to both 'slow' and 'fast' life histories are therefore adaptive in different environmental contexts (*Stearns, 1992*). For example, life-history differences between three amphibian species in Western Europe drove predicted survival and reproduction responses to the North Atlantic Oscillation (*Cayuela et al., 2017*). Generally, organisms with slower life histories are better adapted to cope with environmental fluctuations. Longer-lived organisms have a reduced relative effect of variability in vital rates, variability which is expected during environmental change, on population growth rates (*Morris et al., 2008*) and long-lived plants have weaker absolute demographic responses to weather (*Compagnoni et al., 2021*). However, while generally buffered, long-lived, slow-reproducing animals are often more at risk of extinction (*Cardillo et al., 2005*), and slower to recover when perturbed (*Gamelon et al., 2014*; *Jackson et al., 2019*; *Turkalo et al., 2017*). Comparative approaches linking life-history traits to climate-change responses may therefore provide a useful predictive link to improve our understanding of climate vulnerability.

In this study, we investigated annual population responses to temperature and precipitation anomalies (i.e. weather deviation from average conditions) in populations of terrestrial mammals across the world's ecosystems. We tested whether life history predicts population responses to the weather, and therefore its utility in assessing vulnerability to climate change. We addressed these questions using 486 long-term (≥10 consecutive years) abundance records from 157 species of terrestrial mammal obtained from the Living Planet Database (*Almond et al., 2020*), by implementing a two-step meta-regression framework. First, for each abundance record, we assessed how observed annual population growth rates were influenced by standardised weather anomalies (annual deviation from long-term average weather patterns) and intra-annual weather variance using autoregressive additive models that accounted for temporal autocorrelation in abundance records and overall abundance trends.

Then, we used a phylogenetically controlled Bayesian meta-regression with weather-effect coefficients as the response variable to address three key questions: (1) Are there directional (i.e. different from 0) temperature and precipitation effects on abundance changes across the terrestrial mammals? (2) How are these patterns influenced by covariance both within and between species, and are there vulnerable biomes or spatial patterns in responses? (3) Can species-level life-history traits predict the absolute magnitude of population responses to the weather? We characterised population responses to weather anomalies/variance accounting for both within- and among- species variance, incorporating a categorical predictor of ecological biome (*Olson et al., 2001*), and three broad continuous life-history traits that are widely characteristic of the key axes of life-history variation (*Stearns, 1992*). Due to a lack of overall biodiversity change globally in response to climate (*Antão et al., 2020*), we predicted that there will be no clear directional patterns in population responses to weather anomalies or weather variance overall. Instead, because life-history traits are an evolved response to the environment (*Stearns, 1992*), we predicted that mammals with 'slow' life-history traits will be buffered against weather anomalies and have responses with a lower absolute magnitude (*Morris et al., 2008*). We focused on absolute responses to weather anomalies in the context of life history because we had no a priori expectation for directional patterns in response to weather, but instead expected greater variance in (or more extreme) responses from 'fast' species (*Compagnoni et al., 2021*; *Le Coeur et al., 2021*). We expected that the link between population responses and life history would result in strong phylogenetic autocorrelation in weather responses (*James et al., 2021*; *Melero et al., 2022*). Finally, we predicted that population responses to weather anomalies would be more pronounced in biomes that experience more stable average climatic conditions.

The terrestrial mammals are an ideal study system to explore the predictors of population responses to climate change because they are a well-studied group with a combination of intensive abundance monitoring across the globe (*Almond et al., 2020*), detailed life-history information for hundreds of species (*Conde et al., 2019*; *Myhrvold et al., 2015*) and a highly-resolved phylogeny to facilitate phylogenetic comparative analyses (*Upham et al., 2019*). Furthermore, there is growing evidence from the mammals of the mechanistic links between the climate, demography, and population dynamics (*Coulson et al., 2001*; *Paniw et al., 2019*; *Paniw et al., 2021*; *Woodroffe et al., 2017*).

## Results

We assessed population responses to weather anomalies in 486 long-term abundance time-series records from 157 species of terrestrial mammals globally (*Figure 1*). The time-series records ranged in duration from 10 years to 35 years, with mean and median record lengths across records of 15.7 and 14 years, respectively (*Figure 1*). The records were distributed across 13 terrestrial biomes (*Olson et al., 2001*), including both tropical and temperate regions, but were generally biased towards north western Europe and North America. We had records from 12 of 27 mammalian orders recognised by the IUCN Red List for threatened species (*IUCN, 2016*), but most densely in the Artiodactyla (n=172), Carnivora (n=127) and Rodentia (n=82) (*Figure 1*). The number of records for each species ranged from 1 to 17, with a mean of 3.1 and median of 2 records per species (*Figure 2*).

### No directional population response to weather anomalies

Overall, we did not find directional effects of either temperature or precipitation anomalies on annual population growth rates in the terrestrial mammals (*Figure 2*). In our Bayesian meta-regression, controlling for both within species variance, phylogenetic covariance and differences in record length

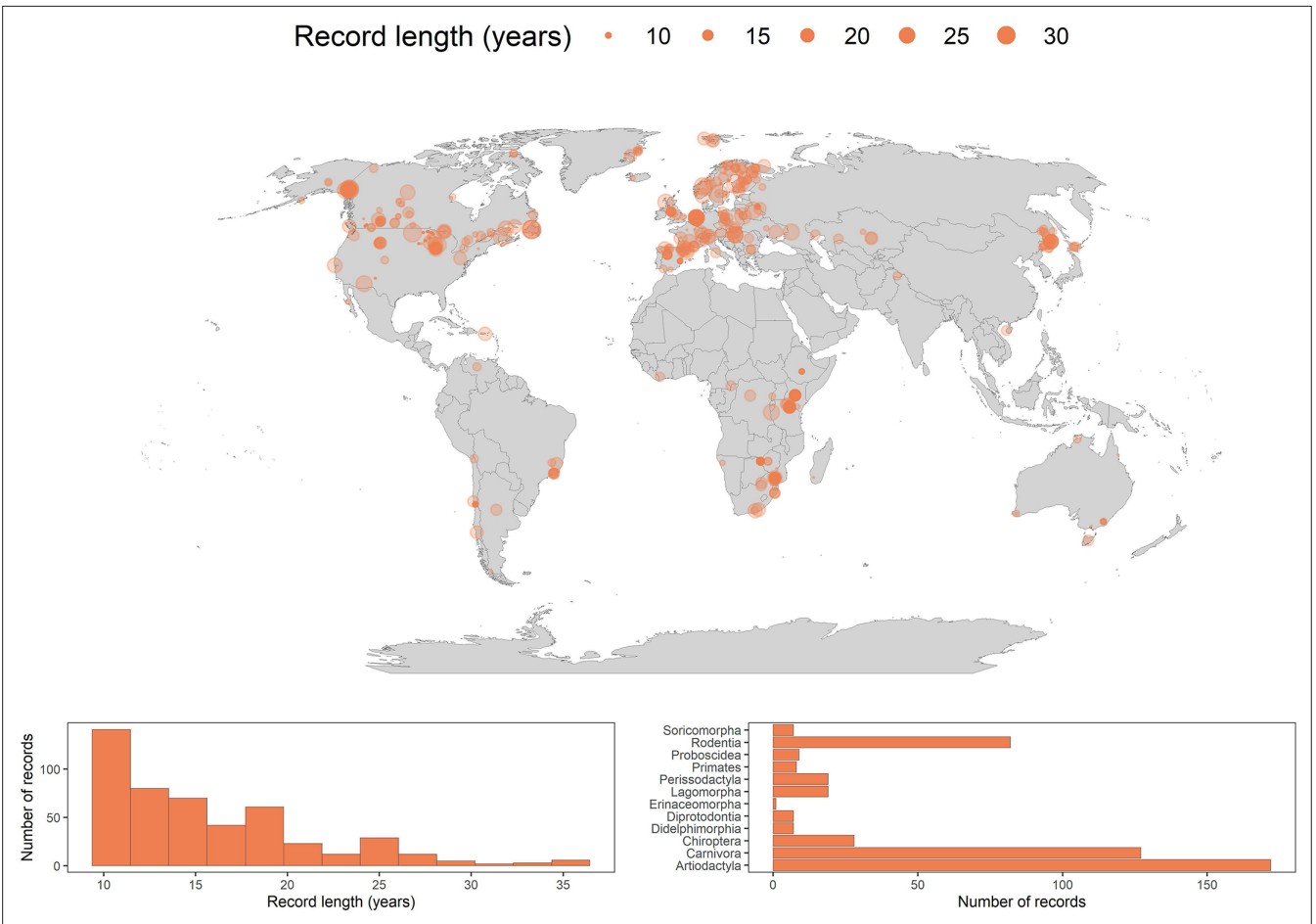

**Figure 1.** 486 long-term abundance records for the terrestrial mammals. Map gives the locations for each record analysed in the current study. Points are transparent, such that colour intensity indicates the spatial density of records. The size of the point represents the record duration in years. The histogram in the bottom left gives the distribution of record lengths across the whole dataset. The bar graph in the bottom right is a frequency distribution of each of the mammal orders analysed in the current study.

(number of years), the posterior mean global intercept, $\bar{\alpha}$, for temperature effects was 0.02 [-0.21,0.25] (95% credible intervals) and for precipitation effects was –0.07 [-0.31,0.15] (*Figure 2a* and *Figure 2b*). Furthermore, 95% of records had temperature and precipitation coefficients from –4.29 to 3.17, and –1.41 to 1.88, respectively. Nevertheless, approximately 8% (n=42) of temperature coefficients and 1% of precipitation coefficients were greater than 3 or less than –3, indicating that small clusters of populations experienced more extreme annual responses to the weather. In addition to temperature and precipitation anomalies, we also found no clear directional effects of weather variance on population growth rates (*Appendix 1—figure 23*). There was also a positive effect of the number of years of population data for a record and the response to temperature anomalies, with a linear slope, $\beta_N$ of 0.12 [0.03,0.21]. Together with the results of the global intercept $\bar{\alpha}$, this suggests that shorter records were associated with more negative temperature effects. However, there was a relationship between the variance in temperature effects and the length of the record; short records displayed larger variation in temperature effects (*Appendix 1—figure 24*). While this finding has important implications for the biases in the raw data, record length was accounted for in all models, and therefore we do not expect that it influences our findings. These results highlight the paradigm of the existence of both winners and losers in weather responses, but no clear directional effect across Mammalia.

## Spatial and phylogenetic effects

We tested whether there were differences in mean weather responses across ecological biomes (*Olson et al., 2001*). We did not find evidence for differences in weather responses across biomes, or strong

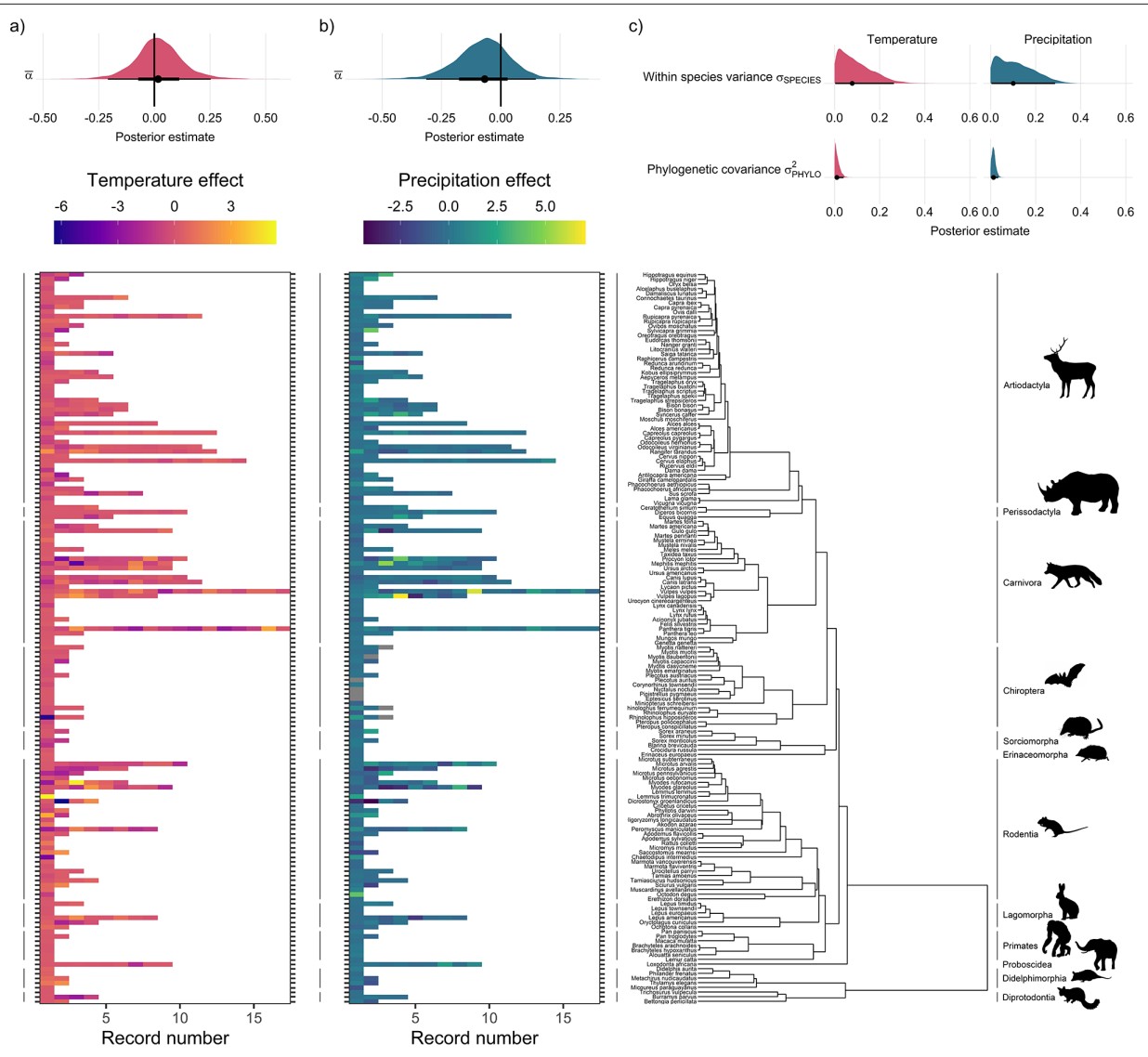

**Figure 2.** Global population responses to weather in the terrestrial mammals. Heatmaps for population responses to temperature (**a**) and precipitation (**b**) for 157 species of terrestrial mammal. Each row of the heatmap is a species, and coloured rectangles are the population records. The colour denotes the coefficient of temperature/precipitation effects derived from autoregressive additive models. Here, positive numbers indicate that positive temperature/precipitation anomalies (hotter/wetter than average in a given year) were associated with increases in population size, and vice versa. The distribution half-eye plots in (**a**) and (**b**) (top) are summaries of the posterior distribution for the global intercept ($\alpha(0)$) of temperature and precipitation responses across records, fit with a Gaussian Bayesian meta-regression. The points give the approximate posterior mean and the error bar is calculated using a cumulative distribution function. Bayesian models were fit incorporating phylogenetic covariance using the maximum clade credibility tree from *Upham et al., 2019*, which is plotted on the right with annotations indicating the mammal order. The distribution half-eye plots in (**c**) are the posterior distribution summaries for phylogenetic covariance and within-species variance from the Gaussian Bayesian meta-regression.

evidence for spatial autocorrelation in weather responses. Using leave-one-out cross-validation, we compared the predictive performance of the model including a categorical predictor of biome relative to the base model, and we found no evidence for increased predictive performance for either temperature ($\Delta$elpd = –0.67; $\Delta$elpd = the change in expected log-wise predictive density relative to the base model) or precipitation ($\Delta$elpd = –0.73) effects (see *Appendix 1—figures 16–17* for more information). Furthermore, we explored the role of spatial autocorrelation in driving differences in weather coefficients across records using Morans I tests and spatially explicit meta-regressions but did not find evidence for spatial autocorrelation in weather effects (*Appendix 1—figures 19–21*).

Interestingly, we found far greater levels of within-species variation in temperature responses compared to among-species variance (*Figure 2c*). To explore the relative effects of within- vs. among-species variance, we incorporated both phylogenetic covariance ($\sigma^2_{PHYLO}$) and species-level variance ($\sigma_{SPECIES}$). The posterior mean for species-level variance in temperature effects was 0.2 [0.01,0.4] which was 20 times greater than the posterior estimate of 0.01 [0.0,0.03] for phylogenetic covariance (*Figure 2c*). Similarly, for precipitation the posterior mean for species-level variance was five times greater than for phylogenetic covariance, with a value 0.05 [0.0,0.15] compared to 0.01 [0.0,0.02] (*Figure 2c*). These patterns are reflected in the temperature and precipitation coefficients, for which large variation can be seen among records of the same species. For example, *Myodes glareolus* (bank vole) in the Rodentia had nine population records, and a range of temperature/precipitation effects from –3.33 to 3.86 and –2.72 to 2.41 respectively, compared to coefficients from –11.60 to 9.22 and –3.47 to 3.22 across Rodentia as a whole (*Figure 2*). This result highlights the potential importance of within-species variability in population responses to environmental change.

## Life history predicts absolute population responses to weather anomalies

Across terrestrial mammals, we found that longer-living species with smaller litter sizes had lower absolute population responses to weather anomalies. We tested a set of Gamma models incorporating univariate, multivariate and two-way interaction effects of maximum longevity, litter size, and adult body mass and their influence on the absolute magnitude of temperature/precipitation effects using model selection and leave-one-out cross-validation (*Supplementary file 1*). As with our Gaussian models of overall weather effects, we found that record length had a strong negative impact on the absolute magnitude of temperature and precipitation responses, with posterior estimates on the linear predictor scale of $\beta_N$ = –0.30 [-0.38—-0.21] and $\beta_N$ = –0.37 [–0.47- –0.26], respectively (*Appendix 1—figure 18*). Namely, shorter records were associated with larger absolute temperature and precipitation responses. We found no association between adult body mass and either temperature ($\beta_{BODYMASS}$ = –0.02 [-0.15,0.10]) or precipitation responses ($\beta_{BODYMASS}$ = –0.00 [-0.17,0.17]). Furthermore, we found no strong evidence for any two-way interactions between life-history variables (*Supplementary file 1*). For both temperature and precipitation effects, the most competitive model was the univariate model including maximum recorded longevity (Δelpd = 5.44 and Δelpd = 1.03, compared to the base model for temperature and precipitation, respectively; *Supplementary file 1* - Table S1). However, univariate models including litter size also had a higher predictive performance than the base model (Δelpd = 3.98 and Δelpd = 0.8 for temperature and precipitation, respectively). For temperature, the second-best predictive model was the one that included univariate effects for longevity, body mass and litter size (Δelpd = 4.54; *Supplementary file 1* - Table S1), and this model was also competitive for precipitation (Δelpd = 0.69; *Supplementary file 1* - Table S2). Therefore, in both cases we selected the models including all univariate life-history effects.

For both temperature and precipitation, our results highlight that shorter living mammals with greater litter sizes had greater absolute responses to weather anomalies than longer-living, slower-reproducing mammals (*Figure 3*). Absolute weather responses were negatively associated with longevity, with posterior means on the linear predictor scale of $\beta_{LONGEVITY}$ = –0.20 [-0.41,0.02] and $\beta_{LONGEVITY}$ = –0.17 [-0.42,0.09] for temperature and precipitation, respectively (*Figure 3a and c*). Thus, a maximum longevity change from 10 months (*Akodon azarae*) to 80 years (*Loxodonta africana*) was associated with a 2.36-fold and 2.05-fold decrease in the predicted absolute magnitude of responses to temperature and precipitation. So, for every additional 5 years of life, there was a 16.8% decrease in absolute responses to temperature and 14.6% decrease in the absolute responses to precipitation. An organism's longevity is strongly correlated to their body mass, but the effect of longevity held irrespective of whether adult body mass was also included in the model.

Absolute weather responses were also positively associated with litter size, with posterior means of $\beta_{LITTER}$ = 0.16 [0.02,0.32] and $\beta_{LITTER}$ = 0.11 [-0.08,0.30] for temperature and precipitation, respectively (*Figure 3b and d*). In other words, mammals bearing more offspring in a single litter had greater absolute responses to temperature and precipitation anomalies. A change in litter size from 1 (monotocous species, various) to 17 (*Thylamys elegans*) was associated with a 1.99-fold and 1.60-fold increase in the predicted temperature and precipitation responses. For every additional offspring

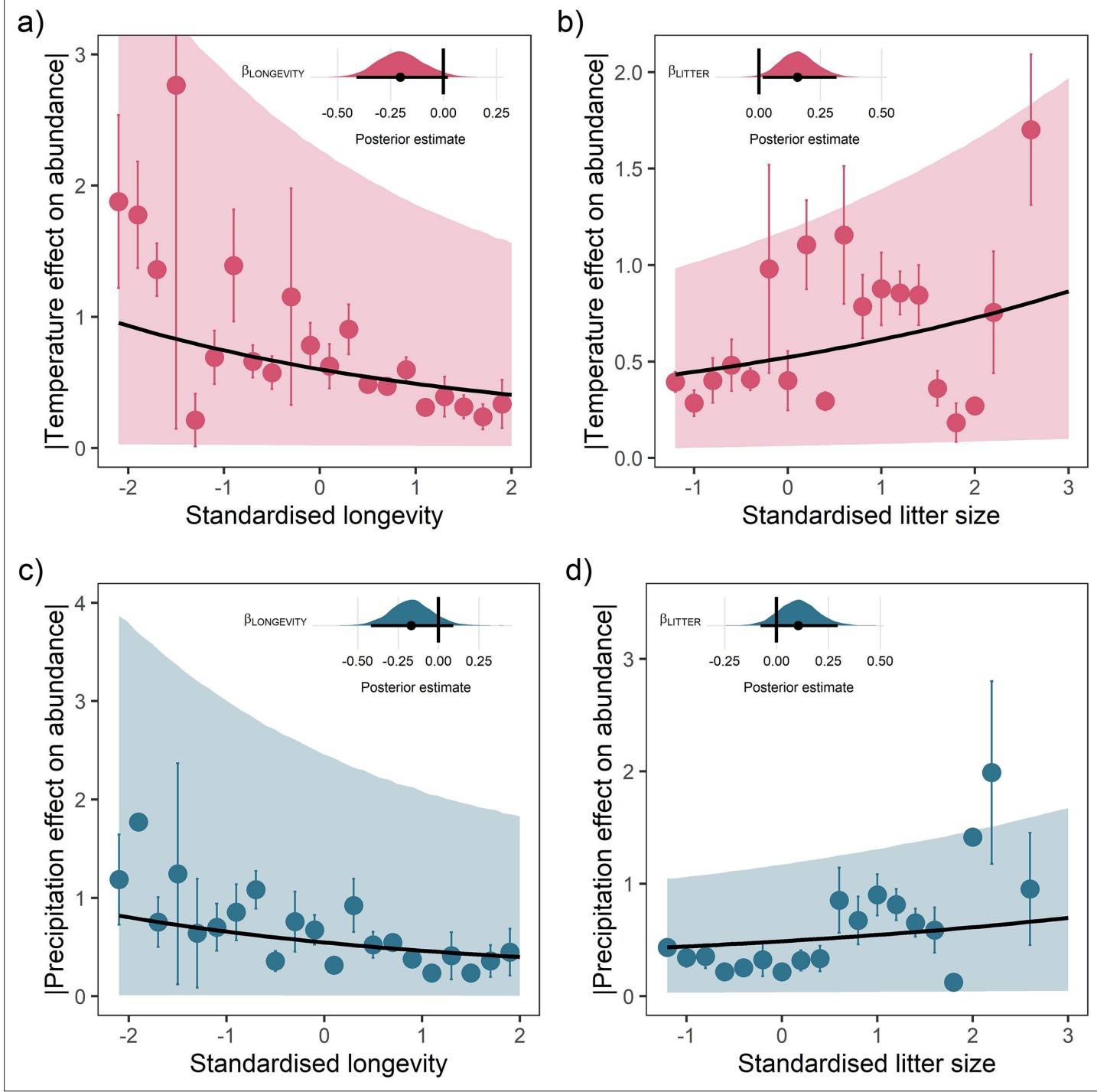

**Figure 3.** Life-history predicts population responses to weather in the terrestrial mammals. Each panel presents the mean absolute effect of temperature (**a** and **b**) and precipitation (**c** and **d**) on population growth rates, $|\omega|$, for standardised maximum longevity (**a** and **c**) and standardised mean litter size (**b** and **d**) (n = 486). Standardisation was performed using z-scores of the natural-log of raw life-history traits. The values on each x-axis are split into equal bins of 0.2 units from the minimum to the maximum life-history value. Points are coefficient means, with standard error bars. The black lines are the mean posterior predictions from the best predictive model, where predictions were calculated averaging over all other covariates and varying effects in the model. The shaded intervals are the 80% quantile prediction intervals. Panel insets give posterior distribution summaries for the slope terms presented in each panel. Two points are omitted from the plotting panel due to large mean coefficient values and high standard errors.

invested into at the litter stage, there was a 12.4% increase in the absolute magnitude of temperature responses and 10% increase in the absolute magnitude of precipitation responses.

## Discussion

Our results provide an important empirical link between a species' life history and its population responses to environmental change. While we found no directional patterns of responses to temperature and precipitation anomalies or intra-annual variance across the mammals, life-history traits relating to the pace of life were associated with absolute responses to weather. Namely, shorter living species with increased litters sizes, or species characterised with 'fast' life-history traits, responded with a greater absolute magnitude compared to those with 'slow' life-history traits. Practically, our results suggest that increased extreme/anomalous weather events will have a greater impact (both positive or negative) on the abundance of short-living mammals with higher reproductive output. Therefore, increased monitoring of vulnerable species with 'fast' life-history characteristics may benefit mammal conservation as global weather anomalies are increasingly common.

We find support for the hypothesis that longevity, and 'slow' life-history traits more generally, buffer organisms against short-term variability in the environment (*Morris et al., 2008*). Life history evolves in response to, and as an adaptation to, environmental conditions (*Stearns, 1992*), but rarely has the link between life history and responses to the environment been demonstrated at a global scale. We add to a small number of studies linking population demography and the climate (*Compagnoni et al., 2021*; *Paniw et al., 2021*). We do not argue that long-lived species are less vulnerable to climate change. Over longer time-scales, species with slow life-history traits are also slower to recover from perturbations (*Gamelon et al., 2014*), which would include sustained environmental change. Furthermore, many long-lived mammal species are affected by other threats such as poaching and habitat loss (*Cardillo et al., 2005*). Critically, however, our results highlight the potential utility of life-history traits for predicting species vulnerability to climate change.

More generally, demography has a role to play in predicting population declines in the Anthropocene and in highlighting targets for conservation management (*Conde et al., 2019*; *Richards et al., 2021*). Our study emphasises this role, demonstrating the predictive power of life-history traits when investigating responses to environmental change. However, there are limitations and barriers to the utility of demography in conservation. Only 1.3% of tetrapods globally have sufficient demographic information with which to estimate population dynamics (*Conde et al., 2019*). Here, we used summary traits that are available for many species (maximum longevity and mean litter size). Although these summary traits were well correlated with more robust demographic traits for a smaller subset of species (*Appendix 1—figure 26*), maximum recorded longevity, while sufficient as a broad indicator, is strongly influenced by sampling variance and a flawed measure of longevity differences between taxa (*Moorad et al., 2012*). Ideally, lifetables with mortality and reproduction trajectories across the life cycle can be combined with data on external drivers to investigate detailed patterns in population dynamics, rather than relying on abundance trends (*Desforges et al., 2018*; *Jackson et al., 2019*). The recent development of the demographic resilience framework, which uses demographic data across the life cycle to simulate how a population may respond to perturbations (*Capdevila et al., 2020*), has excellent potential in extending these findings to explore demographic relationships with climate responses in detail. Unfortunately, however, detailed (st)age-specific demographic information is currently available only for a minority of species, but growing in availability rapidly (*Salguero-Gómez et al., 2016*). Therefore, there is a need to continue to increase the collection of demographic data (and other traits) for many more species than are currently available (*Conde et al., 2019*), so that we may predict population changes with respect to environmental change. Achieving this target may revolutionise the way we quantify species vulnerability to climate change (*Antão et al., 2020*; *Dornelas et al., 2019*; *Leung et al., 2020*; *Paniw et al., 2021*), helping to prevent extinctions before they occur.

In line with recent global assessments of biodiversity in the face of climatic change (*Paniw et al., 2021*), we did not find an overall directional effect of weather anomalies on population growth rates. This may in part reflect the fact that abundance changes are a higher order process determined by complex interactions between demographic components that counteract each other (*Leung et al., 2020*; *Paniw et al., 2021*). However, our results contrast with findings of linear associations between mammal abundance and temperature change (*Spooner et al., 2018*). These differences

may reflect our approach to investigate annual changes, rather than long-term trends. Significant population trends from long time-series are detectable from smaller component time-series even when sampling is incomplete (*Wauchope et al., 2019*), and thus responses detected in trends may reflect broader changes in response to the climate that are not detected in models of annual change. Furthermore, we estimated linear, annual effects of weather on population growth rates, where population responses may actually be more complex non-linear patterns or lagged effects. However, the detection of climate effects on average trends may also be confounded by effects of other (sometimes more dominant) drivers (e.g. habitat loss) (*Daskalova et al., 2020a*). While out of the scope of the current study, population dynamics in endothermic mammals have been linked to other drivers such as diet specialisation, body size and human influence (*Pacifici et al., 2020*). Nevertheless, our findings can be explained in light of recent studies from the Living Planet Database that have found that the large majority of records do not exhibit population declines (*Leung et al., 2020*).

Interestingly, we did not find evidence for phylogenetic covariance in weather responses between species. Closely related species can have disparate responses to weather anomalies. Recent evidence from birds indicated strong phylogenetic covariance in vital rates, particularly in adult survival, and the incorporation of phylogenetic information greatly improved predictive performance when imputing vital rates (*James et al., 2021*). In line with the overall patterns, our findings may reflect the trade-offs between vital rates, which cancel one another out when scaling up to population-level processes such as population growth rates in response to the weather (*Paniw et al., 2021*). However, this result is in contrast to findings from butterflies, which suggested that local responses to weather anomalies had a strong phylogenetic signal (*Melero et al., 2022*). One potential explanation for our finding is that for long-term time-series, there may also be temporal trade-offs in vital rates, where for example investing heavily into survival in one year (in response to weather) may impact subsequent reproduction for several years, decreasing the magnitude of population growth rates. The extent of phylogenetic covariance in vital rate responses and trade-offs remains unknown, but understanding how the climate impacts demographic rates across species may provide a useful tool for imputing population responses to the climate across the tree of life (*James et al., 2021*).

Instead, we highlight the importance of variation in population responses to weather anomalies within a species. Different populations of the same species can have different responses to weather anomalies. Sampling heterogeneity has recently been shown to have broad implications for metrics of population dynamics, where demographic rates are poorly correlated among sampling sites for the same species (*Engbo et al., 2020*; *Römer et al., 2021*). Demographic differences within a species range may reflect broader environmental gradients or wider climatic niche (*Römer et al., 2021*). Therefore, inferences obtained from monitoring single populations or studies may not accurately portray species-level variability (*Gaillard et al., 2013*). This has broad implications for macroecology, particularly in population viability assessments (PVA) and species-distribution modelling. As well as suffering from data quality issues in their parameterisation (*Chaudhary and Oli, 2020*), our findings suggest that PVAs based on data from a single population may not accurately reflect population viability across a species' geographic range. Therefore, incorporating detailed demographic data, and investigating differences in population responses across a range, could greatly improve our perspective on population viability (*Desforges et al., 2018*). Furthermore, presence-only models of species distributions that do not account for responses to the environment within a species range do not accurately represent species distributions (*Benito Garzón et al., 2019*). Moving towards trait-based monitoring and explicitly including demographic processes with mechanistic links to appropriate drivers into species distribution models could greatly improve predictions of climate change impacts on the biosphere (*Trisos et al., 2020*).

## Limitations

As with many macroecological studies exploring global patterns in biodiversity using data collated from many individual studies, it is important to acknowledge limitations in the current study. These limitations include spatial biases, effects of record length on population responses, and the scope of current analyses. First, while broad in coverage spatially, population records in the Living Planet Database are biased towards temperate biomes and Europe/North America, a feature common in macroecology (*Beck et al., 2012*). We aimed to overcome these biases by using data on a well-studied taxonomic group with a broad range of life-history traits, conservative Bayesian meta-regression

incorporating uncertainty, and explicitly exploring spatial and phylogenetic autocorrelation. Second, there was a consistent effect of record length on population responses to weather, for which longer records were associated with lower-absolute ($\omega$ closer to 0) population responses. We accounted for record length in all analyses and do not expect that it influenced our findings, but it is important to acknowledge that increased long-term monitoring is crucial to assess population dynamics. Finally, the scope of the current study was to explore how standardised annual weather anomalies affect the directionality of annual population responses and absolute magnitude of responses with respect to life history. Thus, we do not make conclusions about how life history influences the directionality of responses, nor how long-term climatic trends influence population change.

## Conclusion

Ultimately, improving our predictions of how humans are influencing the natural world is paramount to prevent rapid declines to global biodiversity (*Kissling et al., 2018*). This, however, requires a large shift towards both broad and detailed monitoring of global biodiversity. We show that linking species traits such as life-history traits to changes in the environment may equip us with tools to predict and prevent future losses.

# Materials and methods

To assess the effects of weather anomalies on population growth rates, we collated information on global weather and the abundance, life history and phylogeny of the terrestrial mammals. We carried out all analyses using R version 4.0.5 (*R Development Core Team, 2021*). For all data on the terrestrial mammals, we standardised species names using the *taxize* package version 0.9.98 (*Chamberlain, 2020*) and matched using the Global Biodiversity Information Facility database (https://www.gbif.org/). All code used in the current study and additional descriptions of the analyses are archived in the Zenodo repository (https://doi.org/10.5281/zenodo.6620489), which was created from the following GitHub repository https://github.com/jjackson-eco/mammal_weather_lifehistory; *Jackson, 2022* copy archived at swh:1:rev:cd6fb95ac8ae80c6889fe4f4785d17cab7d18375.

## Time-series abundance data

The Living Planet Database was developed by the World Wildlife Fund and the Zoological Society of London as a tool to monitor global biodiversity, and contains over 20,000 population abundance records for over 4,000 species of vertebrate (*Almond et al., 2020*). We obtained long-term annual time-series abundance data across the terrestrial mammals from the Living Planet Database found at https://livingplanetindex.org/data_portal. The records measure annual abundance in a variety of ways (e.g. full population counts, density, indices). We tested whether the method of data collection influenced the effect of weather on population growth rates, and found that they did not influence overall effect distributions (*Appendix 1—figure 19*). Records contain information on the location, realm, biome, and taxonomy of the species in the record. We included only data for the terrestrial mammals that had species-level life-history information and coordinate locations, which referred to either specific or more general locations for each population (accounted for using weather data from a buffered radius around each location). Our analyses were focussed on estimating weather effects on annual population growth rates using regression models with several covariates, and short timeseries are at large risk of overfitting when including covariates (*Knape and de Valpine, 2011*). Therefore, we opted to include only long-term records with 10 or more consecutive years of abundance data, and only for years in which there was also weather data (1979–2013). We tested the implications of this choice of data quality by running additional analyses with both ≥5 and ≥20 years of data. Our key findings were robust to the choice of long-term records (*Appendix 1—figure 25*), and we therefore continued with 10 years. In one record (for *Bettongia penicillata*), there were two blocks with ≥10 years of data, which were analysed separately. We also removed records (n=8) with a high proportion (>32%) and consecutive occurrences of 0 in the raw abundance time-series (refer to the annual_abundance_changes/ directory, associated README.md and annual_population_growth_rate.R file in the Zenodo repository doi:10.5281/zenodo.6620489). Our final dataset contained 486 records from 157 terrestrial mammal species, which was used in all subsequent analyses (*Figure 1*).

## Global weather data

We used temperature and precipitation anomalies as the key weather variables in our analyses. We extracted global weather data from version 1.2.1 of the CHELSA monthly gridded temperature and precipitation dataset at a spatial resolution of 30 arc seconds (~1 km² at the equator) for all months between 1979 and 2013 across the globe's land surface (*Karger et al., 2017*). We processed raster files of the raw monthly mean temperature and total precipitation data using the *raster*, *rgeos*, and *sf* packages (*Bivand and Rundel, 2020*; *Hijmans, 2020*; *Pebesma, 2018*). For each record, we averaged raw variables for the surrounding region to account for the lack of specificity in record locations and account for animal movements that may alter the weather conditions experienced. Using the Living Planet Database record coordinate locations as a centroid, we averaged the monthly weather data for a buffered radius of 5 km around each record location using the *exactextractr* package (*Baston, 2020*). Averaged weather variables and weather effects for alternate buffer radii (50 m and 50 km) were highly correlated (*Appendix 1—figure 1*; *Appendix 1—figure 6*), and thus we do not expect that our results were sensitive to the choice of this radius.

Generally, given that organisms have evolved in a given environmental context (with seasonal fluctuations), we expect that populations will respond more often to extremes in the weather, as opposed to raw weather changes. Furthermore, across the globes surface the variance in weather variables changes substantially, which may influence population responses. Thus, we explored population responses for the key weather variable of standardised annual anomalies, and then validated our approach using annual weather variance. These weather anomalies are the average distance of the observed temperature and precipitation from expected values in a given year. For the anomalies, we decomposed z-scored averaged monthly weather data for each location for the full timeseries (1979–2013 i.e. a longer timeseries than each record) using a Seasonal-Trend Decomposition by Loess (STL) (*Cleveland et al., 1990*). We refer to z-scoring when variables were mean centered on 0 and standardised by their standard deviation. We used a seasonal window of 7 (seasonal smoothing parameter) and trend window of 1,000 (trend smoothing parameter) for the decomposition (*Cleveland et al., 1990*). We extracted the anomaly component, which describes the remainder when accounting for the trend and seasonal components of the timeseries. We then used annual mean temperature and precipitation anomalies as the key weather variables in subsequent analyses. Weather variance was calculated for each year as the Pearson's variance of monthly mean temperature and monthly total precipitation values.

## Species-level life history

We tested how responses to weather anomalies was associated with a species' position on the 'fast'-'slow' continuum of life history using summary traits. We used three key traits that broadly characterise species-level life history that are available for a large number of species: maximum longevity, litter size and adult body mass. We collected these traits from the compendium developed by *Conde et al., 2019*, combining information from three primary database sources: The Amniote Life-History Database (*Myhrvold et al., 2015*), PanTHERIA (*Jones et al., 2009*) and AnAge (*Tacutu et al., 2013*) databases. Adult body mass data was obtained exclusively from the Amniote Life-History Database (*Myhrvold et al., 2015*). Where multiple records were available for a single species, we took the largest maximum longevity value and the mean litter size/adult body mass. We removed erroneous raw litter size data for *Hydrochoerus hydrochaeris* (mean litter size = 37.8) and *Marmota broweri* (mean litter size = 1063), which both greatly exceeded the 95% quantile for mean litter size (7.13) and contradicted published species information. For analysis, we z-scored the natural-logarithm of raw life-history trait data, and verified that the life-history variables were represented across the range of weather anomaly variables in the raw data (*Appendix 1—figure 2*).

In order to test the suitability of the selected traits for capturing species-level life history, we also explored demographic rates from structured population models and the covariance of life-history traits. We extracted 37 suitable structured matrix population models (namely, ergodic, reducible, primitive, non-NA population matrices) from the COMADRE database (*Salguero-Gómez et al., 2016*). From these matrix population models, we calculated adult survival (mean survival of adult life stages), life-expectancy and generation time. Generally, there was high covariance in all life-history traits, with longevity traits positively associated with adult body mass and negatively associated with litter size (*Appendix 1—figure 26*). The additional life-history traits from structured population models were

also significantly correlated with maximum longevity, litter size and adult body mass (*Appendix 1—figure 26*). Furthermore, we repeated subsequent analyses for 16 species that had both detailed demographic rates and population abundance data, and found that the link between life history and population responses to weather anomalies was maintained (*Appendix 1—figure 27*).

### Phylogeny data

The mammal phylogeny was obtained from *Upham et al., 2019*, which uses a 'backbone-and-patch' Bayesian approach for a newly assembled 31-gene supermatrix and is part of the Vertlife project (https://vertlife.org/). We used the maximum clade credibility tree in analysis, which was processed using the *ape* package (*Paradis and Schliep, 2019*). *Loxodonta cyclotis* (African forest elephant) was considered as *Loxodonta africana* (African elephant) for analysis so that the abundance record and phylogenetic data matched.

### Weather effects on annual population growth rates

To assess comparative population responses to weather anomalies in the terrestrial mammals we used a two-step meta-regression approach. First, for each record we estimated the effect of annual weather anomalies (and weather variance) on population growth rates. We calculated the standardised proportional population growth rate r in year t as

$$r_t = \ln\frac{X_{t-1}}{X_t}, \tag{1}$$

where X is the abundance in year t, transformed (raw abundance +1) to prevent observations of 0. We used this standardised population growth rate to ensure that the effects of weather on population growth rates were on the same scale across the population records (e.g density measured between 1 and 5 individuals per km$^2$ vs. full population counts between 10,000 and 50,000 individuals).

Then, with $r_t$ as the response variable, we estimated the effect of temperature and precipitation anomalies on population growth using generalised additive mixed models (GAMMs) fit using the *gamm* function of the *mgcv* package (*Wood, 2017*). We opted to use a general linear-modelling framework as opposed to a state-space approach, which is often employed for time-series to account for measurement error and estimate trends (see *Daskalova et al., 2020b*). The primary reason for this choice was that we aimed to assess broad comparative patterns in population change, and did not expect systematic errors in model parameters due to measurement error. Furthermore, *Daskalova et al., 2020b* found that abundance trend terms were highly correlated between linear and state-space approaches across the Living Planet Database, which would be expected if there are not systematic errors in measurement across the database. Our results were robust to this modelling choice, and we found that the observed population responses were highly correlated between GAMM and state-space approaches (*Appendix 1—figure 11*; see full alternative approaches section below).

In addition to estimating the influence of weather anomalies, we accounted for temporal autocorrelation in abundance and trends in population change. Changes in abundance are influenced by several drivers of population dynamics including habitat loss (*Daskalova et al., 2020a*) and population processes such as density dependence (*Brook and Bradshaw, 2006*), which may confound any influence of the weather on abundance. Therefore, because we aimed to assess the isolated impact of weather anomalies, accounting for these trends in abundance and temporal autocorrelation was crucial. We initially explored the extent of autocorrelation in abundance patterns using timeseries analysis and found evidence for lag 1 autocorrelation in abundance, but not for greater lags (*Appendix 1—figures 3 and 4*). Furthermore, we tested the potential impact of density dependence on estimating environmental effects using an autoregressive timeseries simulation and found that environmental effects were robust to density dependence even for short timeseries (*Appendix 1—figure 5*).

Thus, to estimate the effect of weather anomalies on population growth, for each record we modelled population growth rate in each year as

$$r_t = \beta^0 + \omega W_t + f(y_t), \tag{2}$$

where $\beta^0$ is the intercept and $\omega W_t$ is a linear parametric term with coefficient $\omega$ for the weather W (temperature or precipitation anomaly) in year $t$. Here, positive coefficients indicate that positive weather anomalies i.e. hotter/wetter years, were associated with population increases, and *vice versa*.

Identical additive regression models were run using weather variances as the weather variable W. The term $f(y_t)$ captures the effect of year $y_t$ as a non-linear trend. Here, the smoothing function $f$ was fit using a thin plate regression spline, which is comprised of penalised local regressions, where the number of regressions is given by the basis dimension (**Wood, 2003**). We used a basis dimension of five. The function $f$ was also fitted with an order 1 autoregressive (AR(1)) correlation structure, as specified in the *nlme* package (**Pinheiro et al., 2014**). Thus, using the year effect we accounted for both a non-linear trend in abundance and temporal autocorrelation.

## Alternative approaches to estimate weather effects

We validated our additive model approach by testing other models to calculate weather effects, including linear regressions both including and excluding temporal trends or density dependence, state-space models and a temporally autocorrelated model fit using the *glmmTMB* package (**Brooks et al., 2017**; *Appendix 1—figures 7–11*). Weather coefficients $\omega$ generated using linear year effects were positively correlated to those from additive models (*Appendix 1—figure 9*). In addition to models fit using a generalised linear modelling approach, we also tested the validity of the final GAMM approach by calculating weather effect coefficients using state-space autoregressive time-series models that incorporated both process and observation error, which are often used as predictive models of time-series abundance data (e.g. **Daskalova et al., 2020b**). Here, the state process of the population growth rate r in year t was a function of a linear effect of the weather variable on the abundance and random noise. An advantage of state-space approaches in this context is that explicitly modelling process noise captures inherent variation in population growth rate that may confound linear relationships with weather anomalies. We fit state-space models using the *rjags* and *jagsUI* packages in R (**Kellner, 2021**; **Plummer, 2019**) across 3 chains, which each had a total of 200,000 iterations, comprised of 100,000 burn-in iterations, 5000 adaptation iterations, and a thinning rate of 6. Across time-series records, there was a high fit-to sample, and the fit-to-sample was not influenced by the length of the time-series record (*Appendix 1—figure 10*). We compared weather coefficients from state-space models with those obtained from GAMMs using Pearson's regression, and found highly significant correlations for both temperature and precipitation effects (*Appendix 1—figure 11*). Overall, given the strong correlations observed between weather variables calculated using different approaches, we concluded that our results are unlikely to be sensitive to the choice of modelling framework.

## Bayesian meta-regression

With the weather effects $\omega$ from each record as the response variable, we explored comparative patterns in population responses to weather anomalies using a Bayesian meta-regression framework implemented in the *brms* package (**Bürkner, 2017**). We standardised weather effects with z-scoring for analyses. We fit separate models for temperature and precipitation. Then, we used Bayesian meta-regression to address three key questions: (1) Were there directional (i.e. average responses different to 0 overall) population responses to weather across the terrestrial mammals? (2) How did population responses vary within and between species and were there spatial patterns across biomes? (3) Does life history predict the absolute magnitude of population responses $|\omega|$? To address questions 1 and 2, we used Gaussian models controlling for both phylogenetic and species-level covariance. The full model for record i and species j is given by *equation 3* below

Linear model

$$\omega \sim \text{MVNormal}(\mu, \mathbf{S})$$

$$\mu_i = \alpha_{SPECIES[j]} + \beta_{BIOME}[i] + \beta_N N_i$$

Varying effects

$$\mathbf{S} = \sigma^2_{PHYLO}\mathbf{V}$$

$$\mu_i = \alpha_{SPECIES[j]} + \beta_{BIOME}[i] + \beta_N N_i$$

(3)

Priors

$$\bar{\alpha} \sim \text{Normal}(0, 0.3)$$

$$\beta_{BIOME} \sim \text{Normal}(0, 0.3), \quad \text{for b in } 1:13$$

$$\beta_N \sim \text{Normal}(0, 0.5)$$

$$\sigma^2_{PHYLO} \sim \text{Exponential}(8)$$

$$\sigma^2_{SPECIES} \sim \text{Exponential}(8)$$

where the weather effect $\omega$, is given by a multivariate normal distribution with mean μ and phylogenetic covariance matrix $\mathbf{S}$. The global intercept is given by $\bar{\alpha}$, which estimates overall patterns in weather effects across records, addressing question 1. We incorporated phylogenetic covariance using a Brownian motion model, with the correlation matrix given by $\mathbf{V}$ (calculated from the maximum clade-credibility tree) and variance factor $\sigma^2_{PHYLO}$, from which between-species variance was estimated. We incorporated an intercept-only varying effect for species with the term $\sigma^2_{SPECIES}[j]$, from which within-species variance was estimated with $\sigma^2_{SPECIES}$. The term $\beta_{BIOME}$ gives the spatial effect of biome on weather responses, where biome was a categorical variable with biomes as described by *Olson et al., 2001* (n=13, subscript b). Therefore, we explored question 2 by capturing within-species variance ($\sigma_{SPECIES}$), between-species variance ($\sigma^2_{PHYLO}$), and the spatial effect of biome ($\beta_{BIOME}$). All meta-regression models also included the linear effect of record length N (scaled number of years in the record) on weather effects, which was estimated using $\beta_N$. Finally, we also fit Gaussian meta-regression models for weather effects calculated using the annual weather variance, and the results obtained were largely identical to those obtained for weather anomalies (*Appendix 1—figure 22*).

## Prior predictive simulation

For all meta-regression models, we used conservative priors that gave predictions lying within the parameter space of the raw data. Specifically, we used regularising priors obtained from prior predictive simulations of the slope, intercept and exponential variance terms (*McElreath, 2020a*; *McElreath, 2020b*; *Appendix 1—figures 12–15*). Here, we compared the estimates and predictions of priors to the limits of observed data and expected patterns to inform the priors. In addition to prior choices made in this section, we further tuned the priors during the model selection to improve the efficiency/accuracy of Markov chains. For example, for Normal priors we tuned parameters with further reductions in standard deviation to improve the efficiency of the Markov chains. Choosing conservative regularising priors was also appropriate given the large number of parameters in phylogenetically or spatially controlled models. We performed prior predictive simulation for the global intercept term of directional weather effects (question 1), the β terms relating to differences in weather effects (i.e. biome effects; question 2), β terms for linear life-history effects (question 3), and mixed-effects variance terms for species variance and phylogenetic covariance (question 2). For the full set of priors used in analyses please refer to the meta_regression/ directory of the supplementary code doi:10.5281/zenodo.6620489.

## Life-history effects on weather responses

For question 3, we tested how species-level life history influences absolute responses to weather anomalies. Although on average we expect that species life history influences population responses to the environment, we have little evidence to suggest that life history per se influences the directionality of responses (*Morris et al., 2008*). Thus, to address this question we explored how maximum longevity, litter size and adult body mass influenced the absolute magnitude of weather responses, |$\omega$|, using Gamma regression models with a log link (*Compagnoni et al., 2021*). The full model for record i and species j is given by *equation 4* below

Linear model

$$|\omega| \sim \text{Gamma}(\eta, \mu)$$

$$\log \mu_i = \alpha_{SPECIES[j]} + \gamma_{PHYLO}[j] + LH + \beta_N N_i$$

Varying effects

$$\alpha_j \sim \text{Normal}(\bar{\alpha}, \sigma_{SPECIES})$$

$$\gamma_j \sim \text{MVNormal}(0, \mathbf{S})\mathbf{V}$$

$$\mathbf{S} = \sigma_{PHYLO}^2 \mathbf{V}$$

(4)

Priors

$$\bar{\alpha} \sim \text{Normal}(0, 0.3)$$

$$\beta_{LH} \sim \text{Normal}(0, 0.2)$$

$$\beta_N \sim \text{Normal}(0, 0.2)$$

$$\sigma_{PHYLO}^2 \sim \text{Exponential}(11)$$

$$\sigma_{SPECIES}^2 \sim \text{Exponential}(8)$$

$$\eta \sim \text{Gamma}(2, 0.6)$$

where $\eta$ is a shape parameter that was fit with a Gamma prior, and LH refers to a set of linear life-history terms ($\beta_1 x_1 + \cdots + \beta_k x_k$) that were explored using model selection. Specifically, for the three life-history traits, we explored a set of models incorporating univariate, multivariate and 2-way interaction terms, as well as a base model excluding all life-history effects. For the full set of 10 candidate models please refer to the supplementary information (*Supplementary file 1*). We fit all life-history effects using the same Normal prior, with mean 0 and standard deviation 0.3. A standard deviation of 0.3 was chosen to improve the accuracy of the Markov chains, after initial regularising values of 0.5 were further reduced (*Appendix 1—figure 14*).

## Model predictive performance

We assessed the predictive performance of candidate models using leave-one-out cross-validation implemented in the *loo* package (*Vehtari et al., 2016*). Models were compared using the Bayesian LOO estimate of out-of-sample predictive performance, or the expected log pointwise predictive density (elpd)(*Vehtari et al., 2016*). All final meta-regression models were run over 3 Markov chains, with 4,000 total iterations and 2000 warmup iterations per chain. Model convergence was assessed by inspecting Markov chains and using $\hat{R}$, which assesses the degree of mixing (agreement) between- and within- chains, such that values of $\hat{R} < 1.05$ indicate sufficient agreement across chains.

## Acknowledgements

We extend a warm thank you to the Living Planet Index team from the Zoological Society of London and the World Wildlife Foundation for the opportunity to work with this rich data source, and all the contributors to this amazing resource. We also thank Dylan Z Childs for his support on modelling the impact of density dependence on the abundance timeseries. We thank Rob Goodsell for advice on the Bayesian modelling framework, Christopher Cooney for advice on the mammal phylogeny and the Vertlife project, and Morgane Tidière for stimulating discussion and advice on the project. Thank you also to Dalia Conde and Johanna Stärk for their help with the demographic data from Conde et al. (*Conde et al., 2019*), and the members of the Interdisciplinary Centre for Population Dynamics (CPop), and in particular Jim Oeppen, for useful feedback on the methodology. This work was supported by the Danish Independent Research Fund (DFF-6108-00467B).

# Additional information

### Funding

| Funder | Grant reference number | Author |
|--------|------------------------|--------|
| Danish Independent Research Fund | DFF-6108-00467B | John Jackson Owen Jones |

The funders had no role in study design, data collection and interpretation, or the decision to submit the work for publication.

### Author contributions

John Jackson, Conceptualization, Data curation, Formal analysis, Investigation, Methodology, Validation, Visualization, Writing - original draft, Writing – review and editing; Christie Le Coeur, Formal analysis, Investigation, Methodology, Writing – review and editing; Owen Jones, Conceptualization, Funding acquisition, Methodology, Project administration, Resources, Supervision, Writing – review and editing

### Author ORCIDs

John Jackson (ID) http://orcid.org/0000-0002-4563-2840
Christie Le Coeur (ID) http://orcid.org/0000-0002-0911-2506

### Decision letter and Author response

Decision letter https://doi.org/10.7554/eLife.74161.sa1
Author response https://doi.org/10.7554/eLife.74161.sa2

# Additional files

### Supplementary files

• Supplementary file 1. Model selection results tables for effects of life-history on population responses to weather anomalies in terrestrial mammals.

• Transparent reporting form

### Data availability

All data presented in the current manuscript is publicly available. All code and analyses are fully available and archived in the following Zenodo repository: https://doi.org/10.5281/zenodo.6620489, which was created from the folowing github repository: https://github.com/jjackson-eco/mammal_weather_lifehistory, copy archived at swh:1:rev:cd6fb95ac8ae80c6889fe4f4785d17cab7d18375.

The following dataset was generated:

| Author(s) | Year | Dataset title | Dataset URL | Database and Identifier |
|-----------|------|---------------|-------------|------------------------|
| Jackson J | 2022 | jjackson-eco/mammal_weather_lifehistory: Peer review update 3 | https://doi.org/10.5281/zenodo.6620489 | Zenodo, 10.5281/zenodo.6620489 |

The following previously published datasets were used:

| Author(s) | Year | Dataset title | Dataset URL | Database and Identifier |
|-----------|------|---------------|-------------|------------------------|
| Almond R, Grooten M, Petersen T | 2020 | Living Planet Report 2020 | https://livingplanetindex.org/data_portal | LPD, data_portal |
| Conde DA | 2019 | Data gaps and opportunities for comparative and conservation biology | https://doi.org/10.5061/dryad.nq02fm3 | Dryad Digital Repository, 10.5061/dryad.nq02fm3 |

Continued

| Author(s) | Year | Dataset title | Dataset URL | Database and Identifier |
|---|---|---|---|---|
| Myhrvold NP, Baldridge E, Chan B, Sivam D, Freeman DL, Morgan Ernest SK | 2015 | An amniote life-history database to perform comparative analyses with birds, mammals, and reptiles | http://esapubs.org/archive | ALHD, esapubs |

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

## Appendix 1

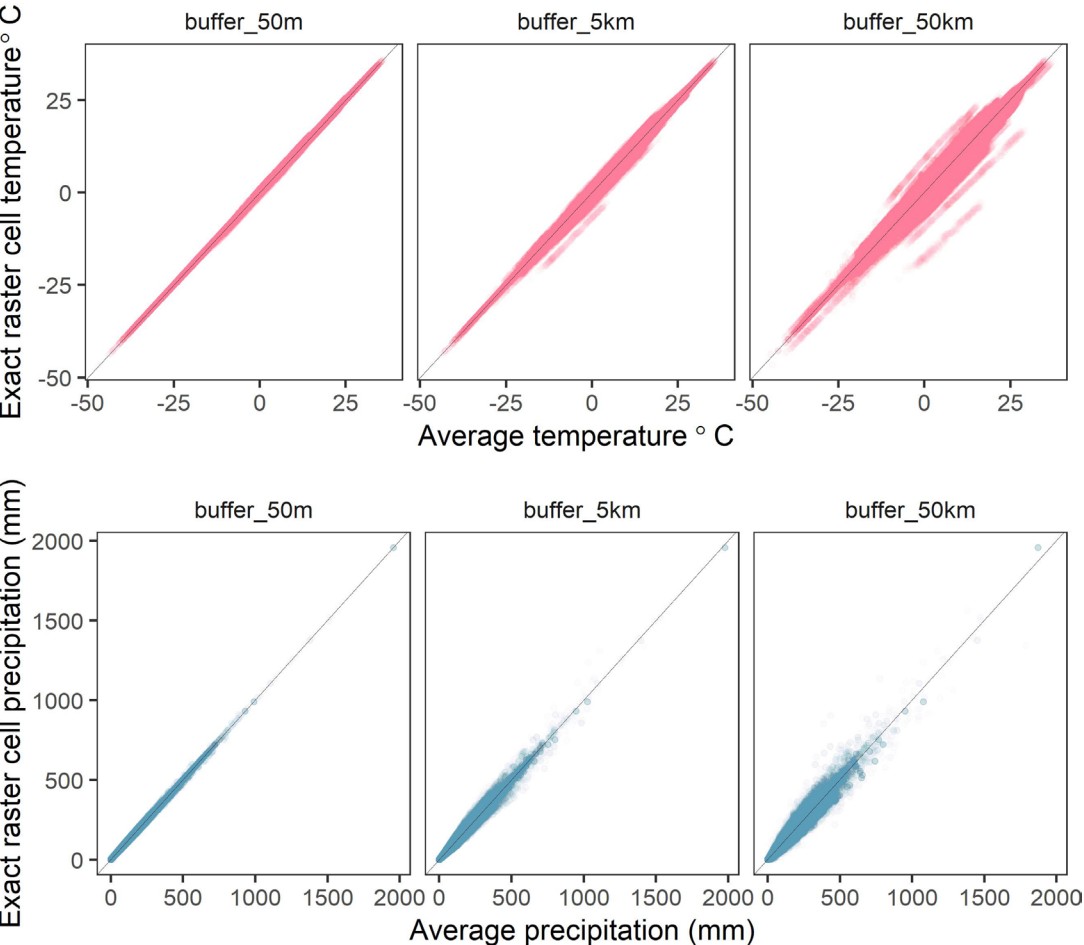

**Appendix 1—figure 1.** Correlations between raster cell weather values for different buffer radii. The average values of mean monthly temperature (top) and total precipitation (bottom) compared to exact raster cell values for buffer radii of 50m-50km (left-right) calculated from the CHELSA global gridded raster dataset. Buffered radii calculated using the *exactextractr* R package.

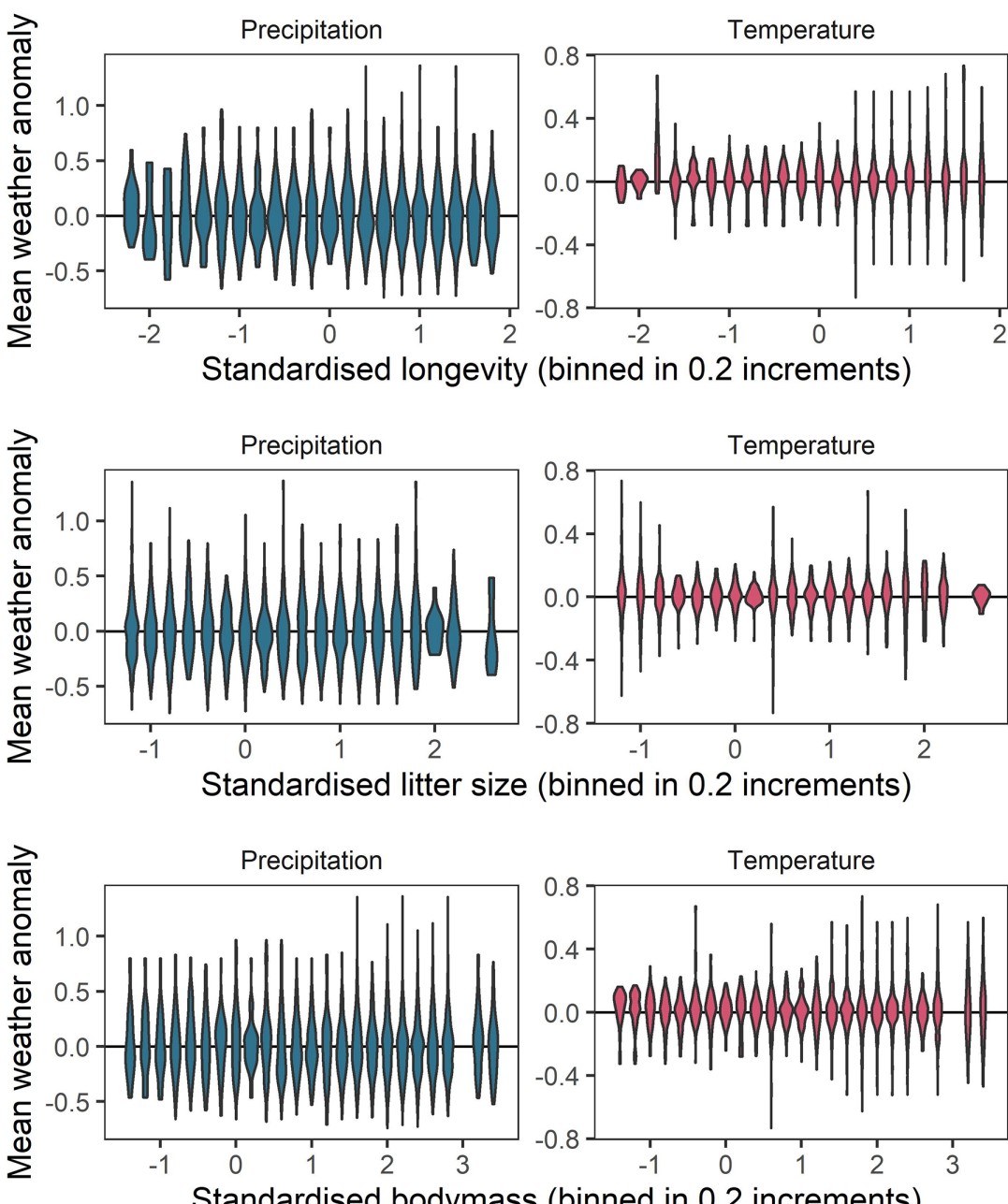

**Appendix 1—figure 2.** Representation of the raw weather anomaly data across the observed ranges of life-history variables. To verify whether a full range of weather anomalies was represented across the range of life-history variables observed, we examined the distributions of weather anomaly values observed for life-history variable bins of 0.2. These panels give violin distributions of weather anomalies for 0.2 increments of each of the three life-history variables (top–bottom) for temperature (right) and precipitation (left). These panels indicate a good spread of weather anomaly values observed across the life-history trait space.

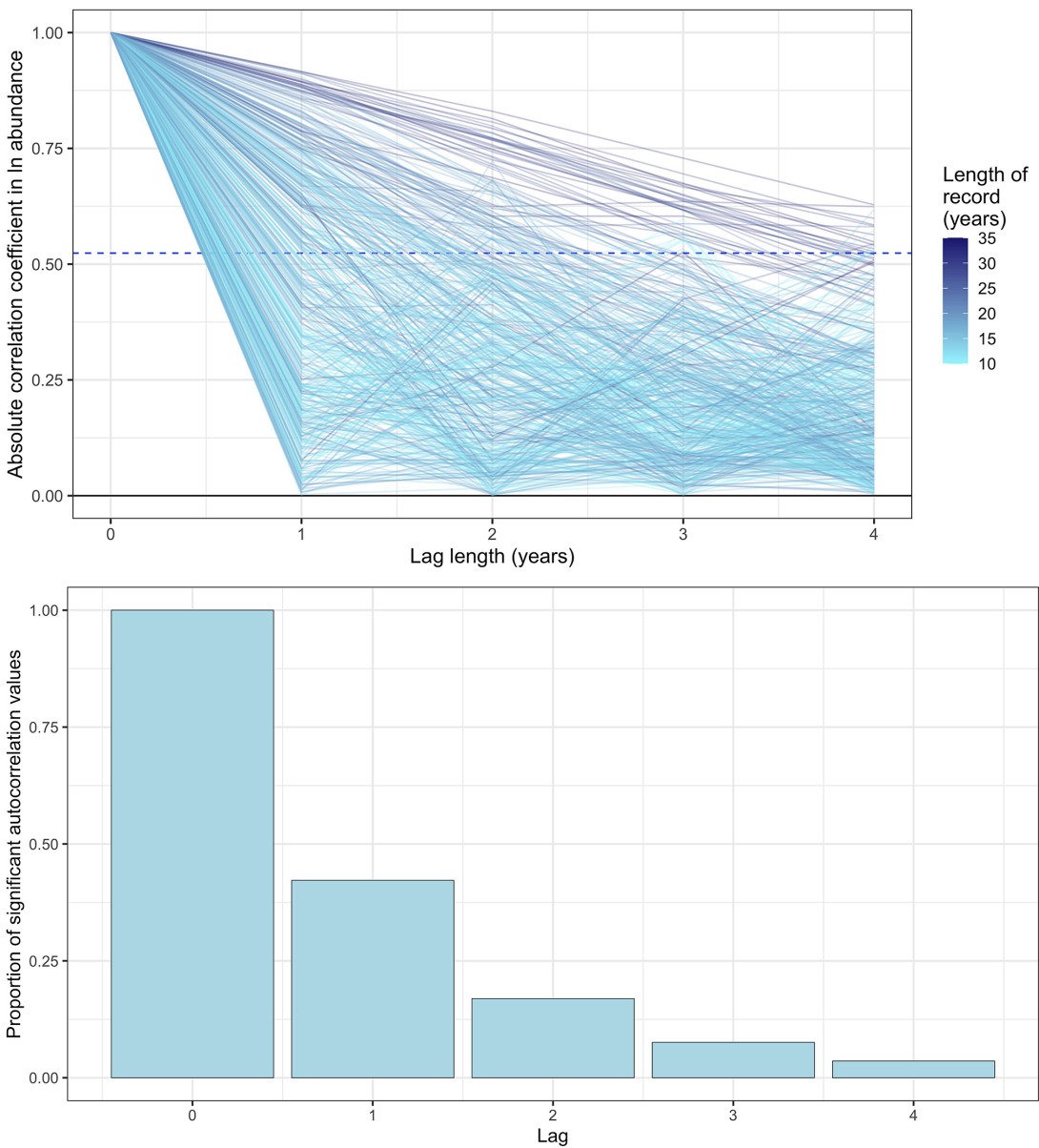

**Appendix 1—figure 3.** Exploring temporal autocorrelation using autoregressive timeseries models of abundance with varying lag. Top- the absolute autocorrelation coefficient for each lag length (in years) for each timeseries abundance record. Each line is a record from the mammal dataset used in the study, with lagged autocorrelation values up to a maximum lag of four years. The colour of the line indicates the total length of the record in years. The dashed line is the significance confidence level for the median timeseries length in the dataset (14 years). Bottom- the proportion of significant autocorrelation values across all records for each lag length (in years). Together, these figures show that there is good evidence for lag 1 autocorrelation (AR(1)) across all records, with a substantial proportion (>30%) of records displaying significant autocorrelation for AR(1). However, with greater lags, the degree of temporal autocorrelation decreases substantially, most probably due to the lack of sufficient annual observations to resolve autocorrelation with a greater lag.

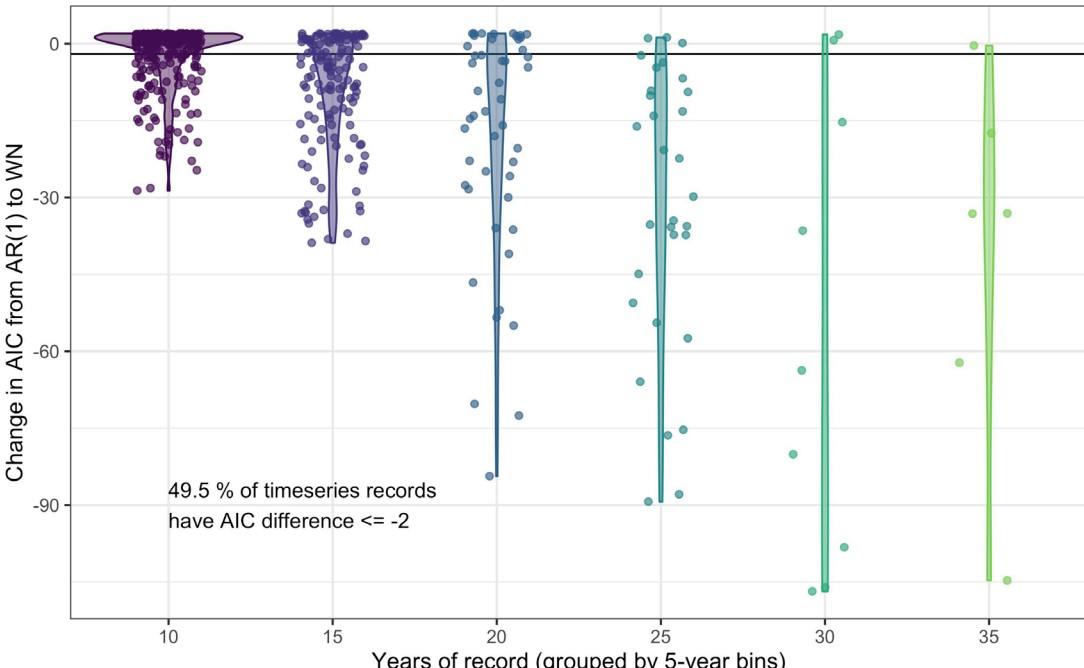

**Appendix 1—figure 4.** Comparing the predictive performance of AR(1) time series models to white noise models for abundance records. The change in AIC for timeseries models including an AR(1) temporal autocorrelation structure relative to the base model of white noise (WN). Each point gives the AIC difference for a single record, with the data grouped by the number of years (bins of five years) in the abundance record. Violins give the distribution of AIC differences for each record length bin. 49.5% of studies have an AIC difference <= −2 when comparing an AR(1) model to a white noise model, indicating support for including lag-1 temporal autocorrelation in models of abundance.

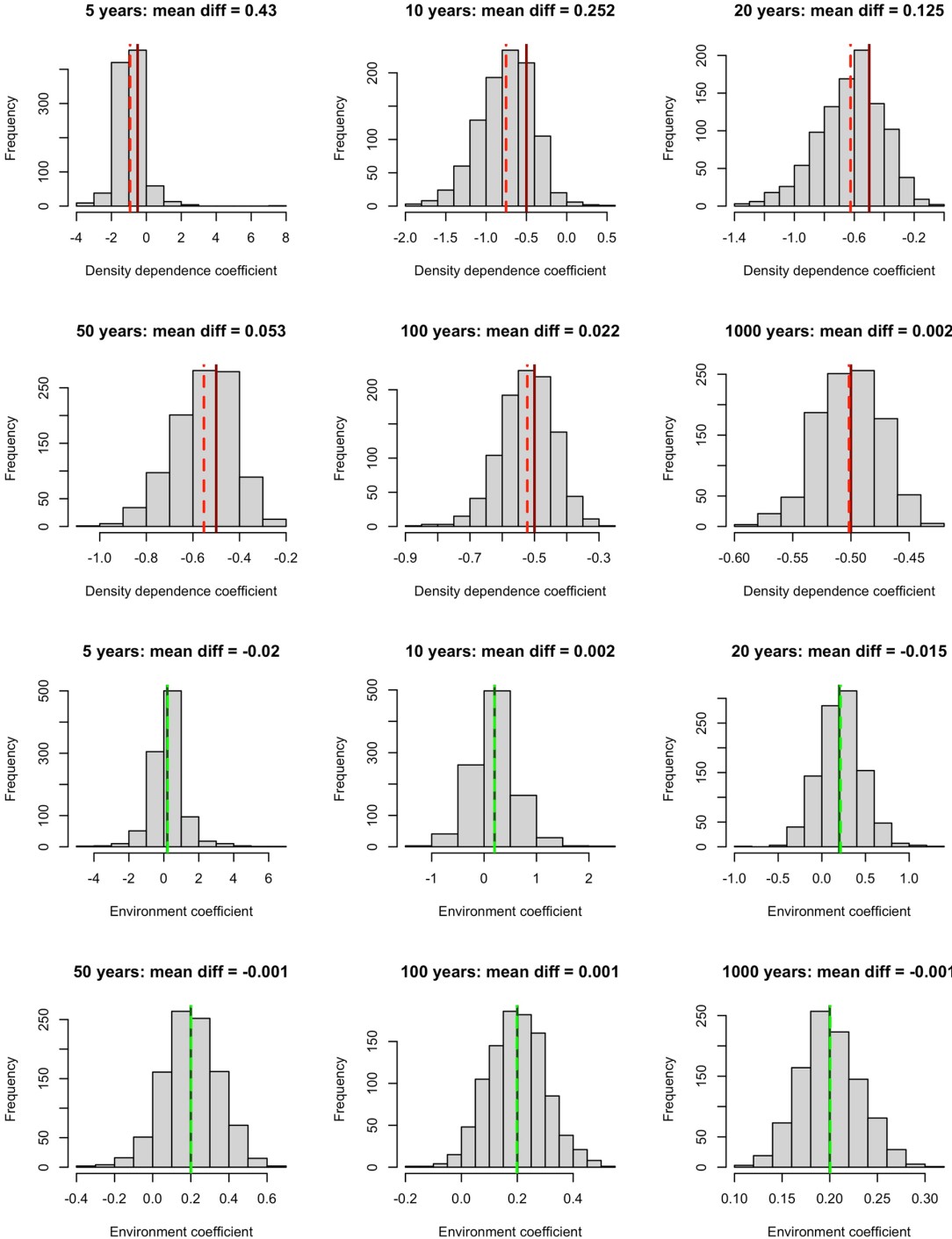

**Appendix 1—figure 5.** Simulation results exploring the impact of incorporating density dependence on the estimation of environmental effects using linear models. We built a time-series model simulation in which there was both lag-1 density dependence (coefficient = –0.5, solid red lines) and an environmental effect (coefficient = 0.2, solid green lines) and then retrofit linear models over 1,000 simulations to explore whether we could estimate density dependence and environmental effects accurately for different time-series lengths. Top 6 panels (red lines) – the distribution of density dependence effects from retrofit linear models (mean = dashed red lines) over 1,000 simulations for increasing timeseries lengths. Here, we did not accurately estimate density dependence effects for short timeseries. Bottom 6 panels (green lines) – the distribution of environmental effects from retrofit linear models (mean = dashed green lines) over 1,000 simulations for increasing timeseries lengths. Here, we accurately estimated environmental effects even for short time series. This suggests that accounting for temporal autocorrelation in abundance, we are able to retrieve accurate environmental effects (here weather effects).

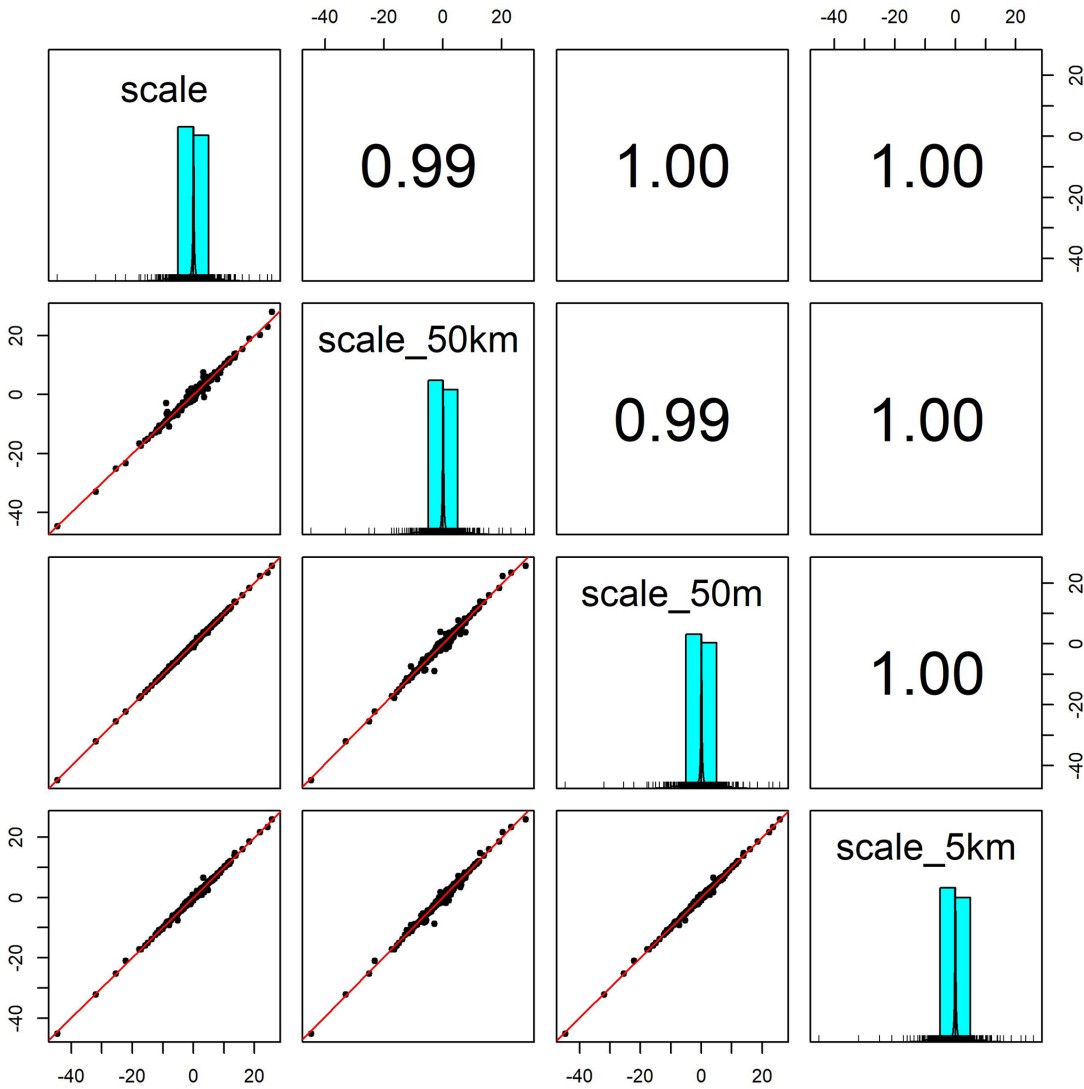

**Appendix 1—figure 6.** Pairwise correlation coefficients for weather coefficients (temperature and precipitation) estimated over four buffered radii scales. We estimated the effect of weather anomalies on population growth rate using additive models (GAMs) over four spatial scales for a buffered radius around each record's location- exact raster cell location (scale), 50m buffer (scale_50m), 5km buffer (scale_5km) and 50 km buffer (scale_40 km). Weather coefficients were near identical across spatial scales. The buffer radius of 5 km was used in subsequent analyses.

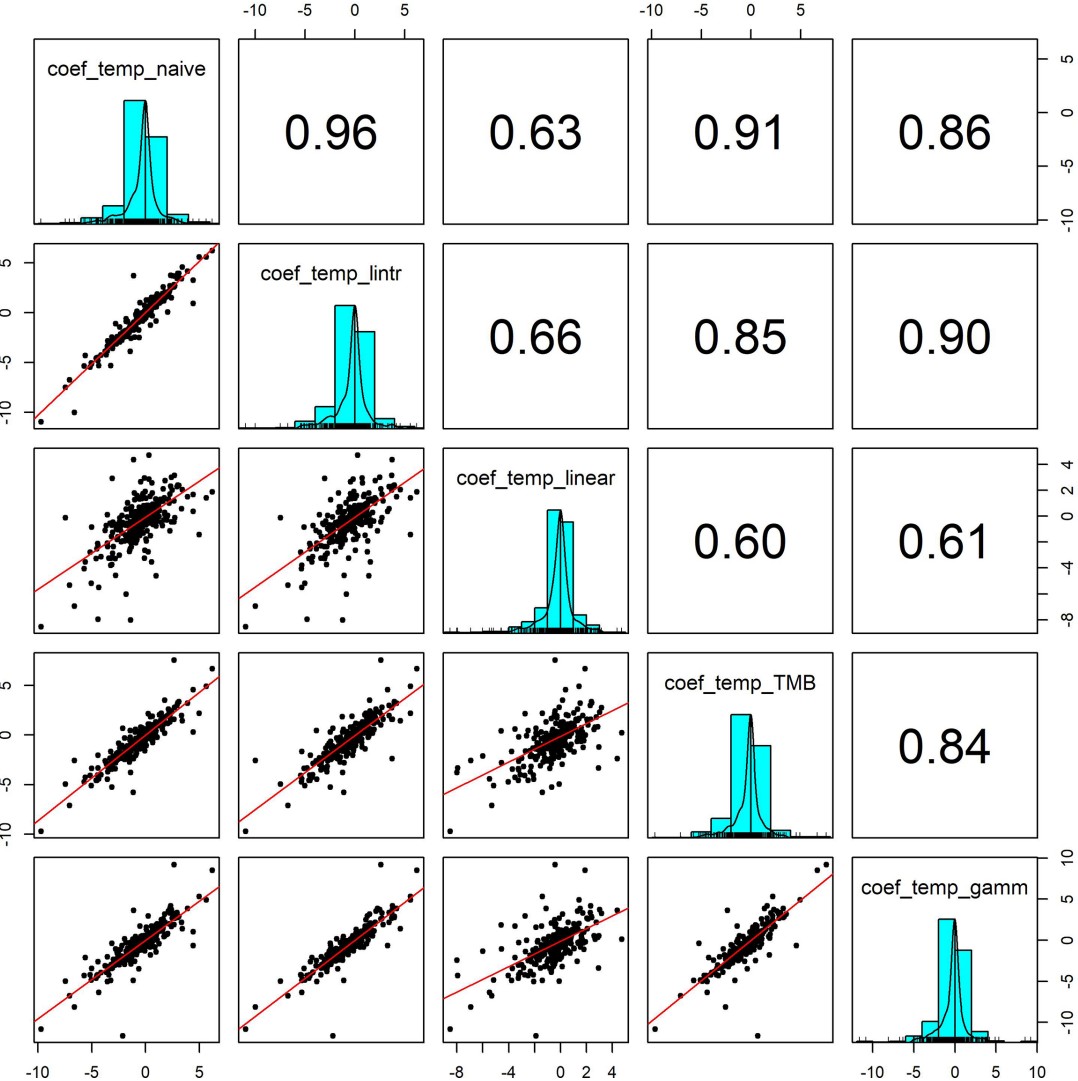

**Appendix 1—figure 7.** Pairwise correlation coefficients for temperature effects estimated using 5 competing generalised linear models. To validate our approach of using additive models to estimate weather effects, we compared the weather coefficients obtained from 5 models of differing complexity that accounted for (or excluded) temporal autocorrelation and temporal trends in population growth rates: (1) coef_temp_naive - simple linear regression excluding temporal trends or autocorrelation (R syntax population_growth_rate ~weather_anomaly), (2) coef_temp_lintr – linear regression including a linear temporal trend in population growth rate but excluding autocorrelation (R syntax population_growth_rate ~weather_anomaly +year), (3) coef_temp_linear – Linear regression including linear trend and an autoregressive term (R syntax population_growth_rate ~weather_anomaly +year + abundance), (4) coef_temp_TMB - A glmmTMB model including an AR(1) autoregressive term for the observation year, (5) coef_temp_gamm – an additive model including a coarse smoothing spline for the temporal trend and an autoregressive term for year (see *equation 2*). Additive model coefficients were highly correlated with other estimates of weather effects (in support of *Appendix 1—figure 5*).

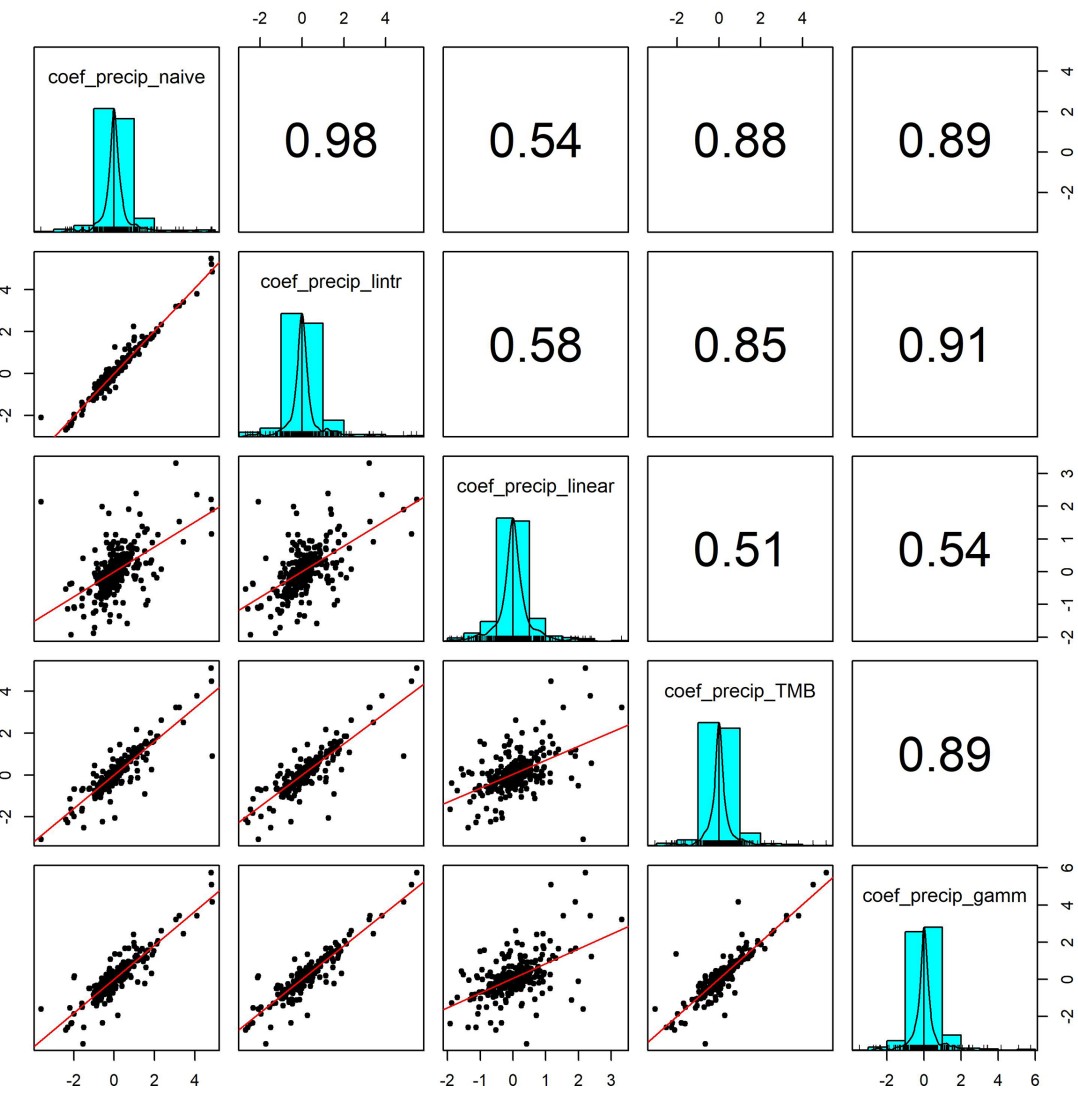

**Appendix 1—figure 8.** Pairwise correlation coefficients for precipitation effects estimated using 5 competing generalised linear models. To validate our approach of using additive models to estimate weather effects, we compared the weather coefficients obtained from 5 models of differing complexity that accounted for (or excluded) temporal autocorrelation and temporal trends in population growth rates (identical to *Appendix 1—figure 7* but using precipitation). Additive model coefficients for precipitation were highly correlated with other estimates of weather effects (in support of *Appendix 1—figure 5*).

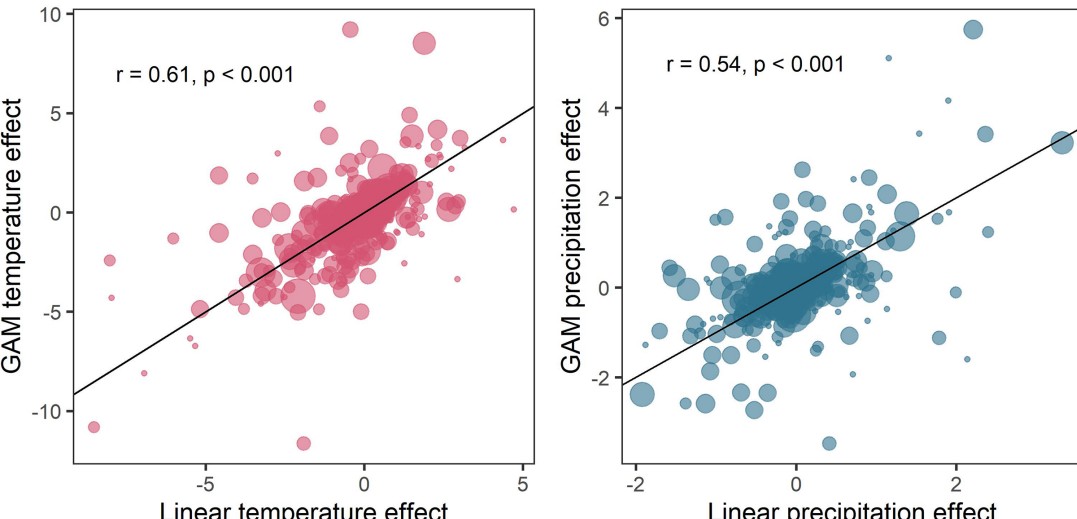

**Appendix 1—figure 9.** Weather coefficients generated from linear regressions were positively correlated to those from additive models. Comparison of weather coefficients generated from linear models including a linear trend term and an autoregressive term to additive models (*equation 2*). Highly significant positive correlation between linear and additive coefficients. This result is in support of the findings of the simulation presented in *Appendix 1—figure 5*, which suggests that despite the method of accounting for density dependence (temporal autocorrelation), estimating annual environmental effects remains robust.

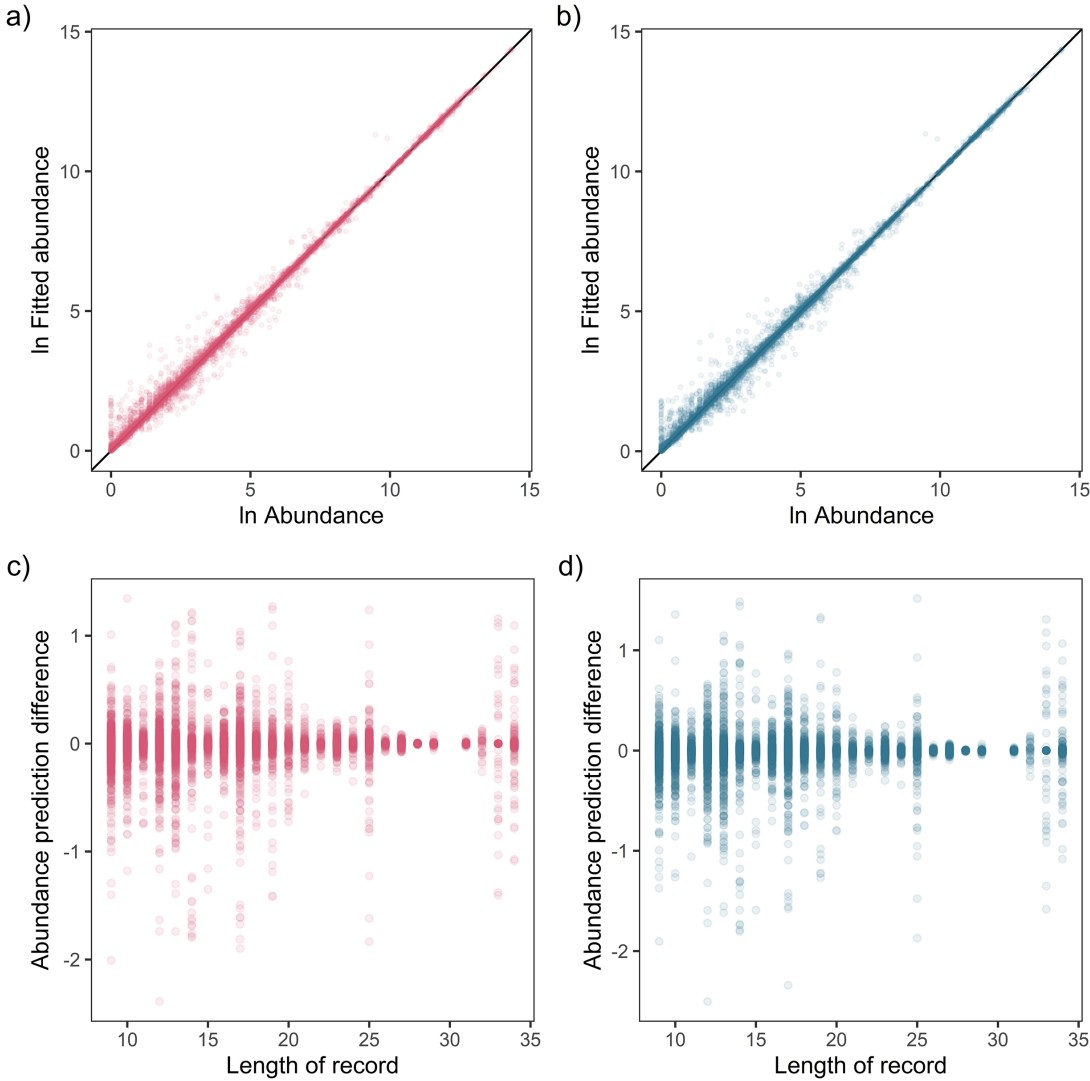

**Appendix 1—figure 10.** Fit-to-sample estimates for state-space models on mammal abundance time-series. Top – Fitted vs. observed annual ln abundance values for temperature (**a**) and precipitation (**b**) effects across all observations of 474 (non-NA in precipitation anomaly) records, with solid black line giving the 1-to-1 line. Bottom – The relationship between the difference in observed and fitted ln abundance with respect to the length of the time-series record for temperature (**c**) and precipitation (**d**).

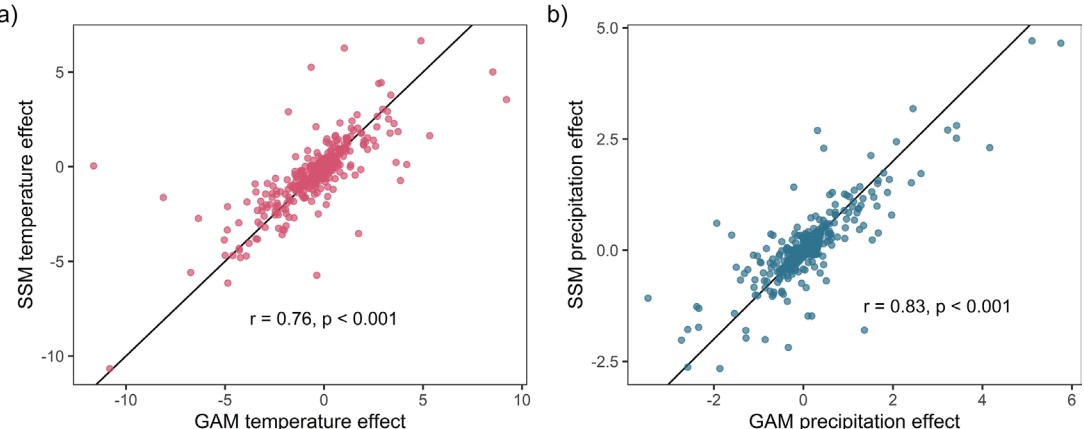

**Appendix 1—figure 11.** Weather coefficients generated from state-space models (SSMs) were highly correlated to those from GAM models. Highly significant positive Pearson's correlation between state-space and additive coefficients for both temperature (**a**) and precipitation (**b**). This result supports the validity of the use of GAM models to estimate weather effects in the current study.

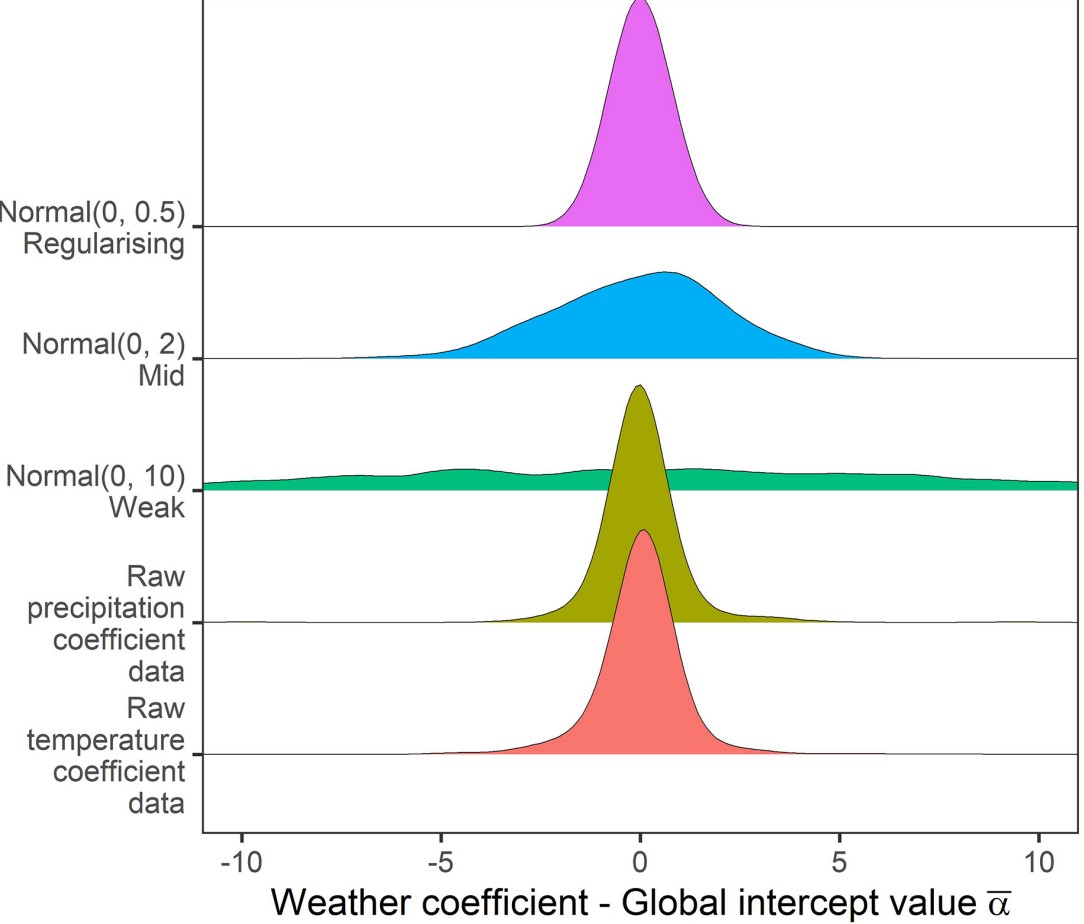

**Appendix 1—figure 12.** Prior predictive simulation for global intercept terms. The global intercept i.e. consistent pattern of weather effects across records was modelled using a normal prior. Density distributions for the weather coefficients observed in the raw data, and weather coefficients under 3 normal priors, weak (mean = 0, sd = 10), medium (mean = 0, sd = 2), regularising (mean = 0, sd = 0.5). Here, the regularising prior gives likely global intercepts within the range of the observed coefficients.

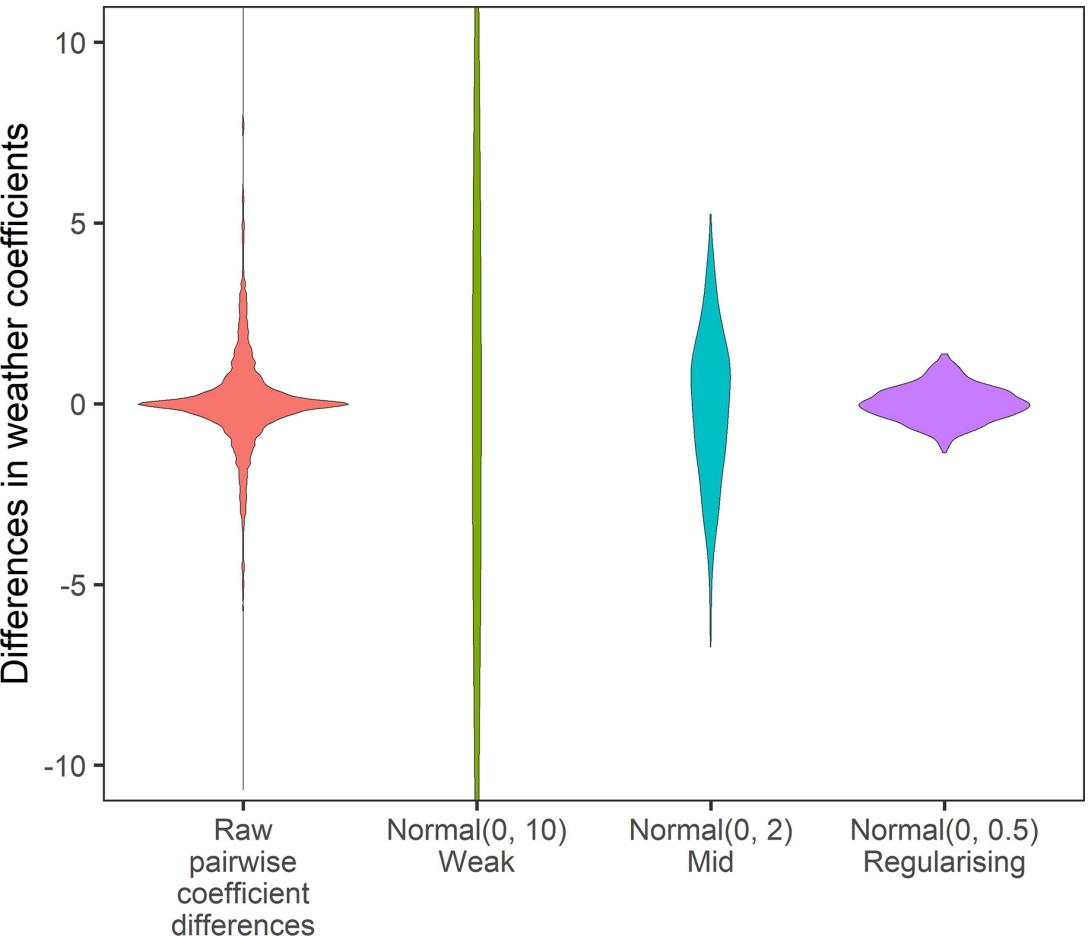

**Appendix 1—figure 13.** Prior predictive simulation for β terms giving differences in weather coefficients. Density distributions for all pairwise differences in the weather coefficients observed in the raw data, and differences in weather coefficients under 3 normal priors, weak (mean = 0, sd = 10), medium (mean = 0, sd = 2), regularising (mean = 0, sd = 0.5). The regularising prior gives difference values within the range of the observed coefficient differences.

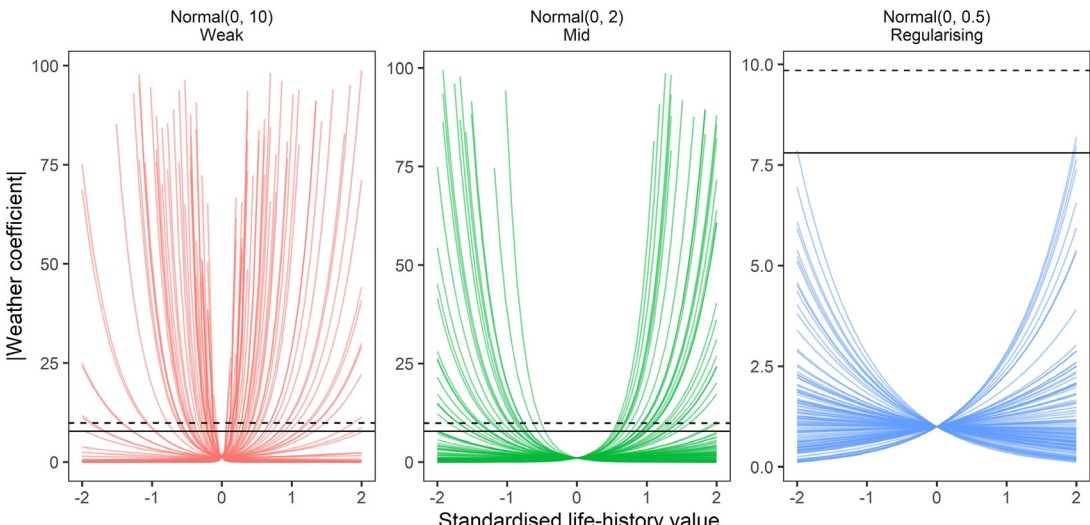

**Appendix 1—figure 14.** Prior predictive simulation for β terms giving life-history effects on weather coefficients. Prior predictions of the effect of simulated scaled life-history values (−2–2) on absolute weather
*Appendix 1—figure 14 continued on next page*

*Appendix 1—figure 14 continued*

coefficients as using log-normal models. Lines are individual simulations under normal prior distributions for the linear life-history effect, which was back-transformed using the exponential to give absolute weather coefficients. Panels give the prior simulations of life-history on weather coefficients under 3 normal priors, weak (mean = 0, sd = 10), medium (mean = 0, sd = 2), regularising (mean = 0, sd = 0.5). solid and dashed horizontal lines give the maximum observed absolute coefficients for temperature and precipitation, respectively. The regularising prior gives plausible predictions that do not regularly exceed the maximum and minimum effects observed.

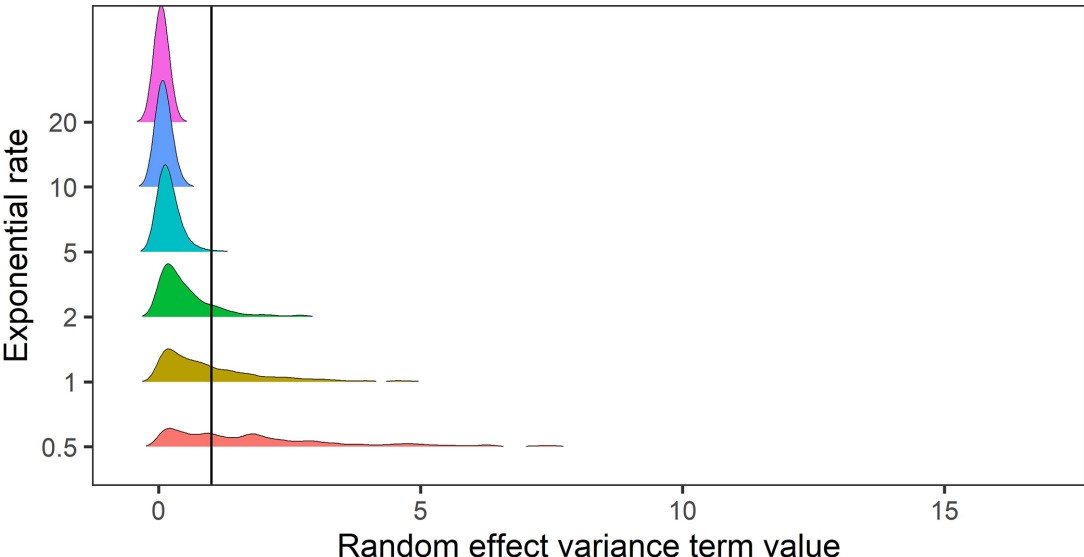

**Appendix 1—figure 15.** Prior predictive simulation for standard deviation terms relating to random effects of variance. The random effects of species-level variance and phylogenetic covariance were modelled using exponential priors, which are suitable for variance terms because they are non-zero distributions that can flexibly capture higher variances. Here, we explored the density distributions of exponential priors with 6 exponential rate parameters (0.5–20). In this case, for phylogenetic and species level variance we do not expect values exceeding a variance term of 1 (solid black line). Regularising priors with rate parameters >= 5 gave conservative estimates of random effect variances within the constraints of the variance terms in the meta-regression models.

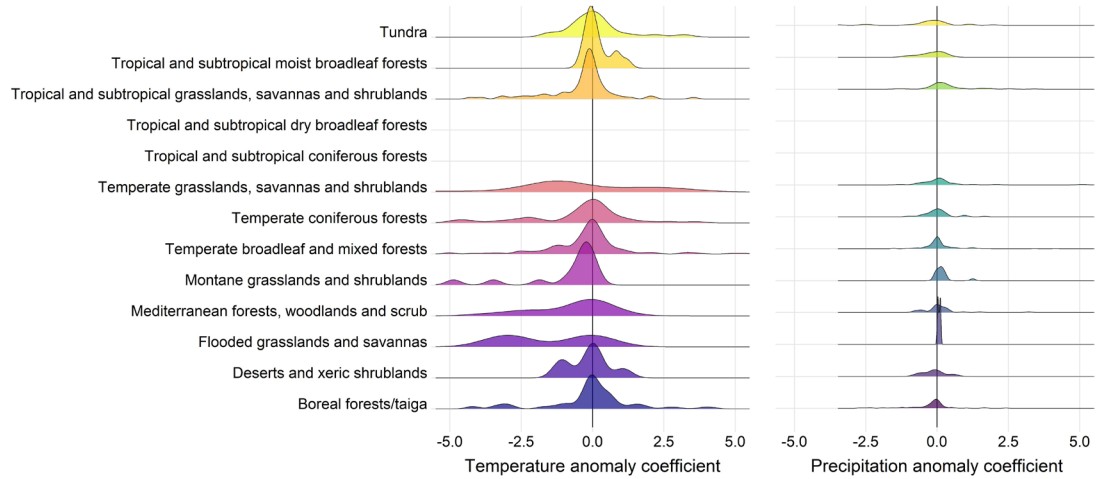

**Appendix 1—figure 16.** Density distributions of temperature (left) and precipitation (right) coefficients on abundance change with respect to biome of the record location in the terrestrial mammals. Only coefficients between –0.5-0.5 are displayed for visual purposes.

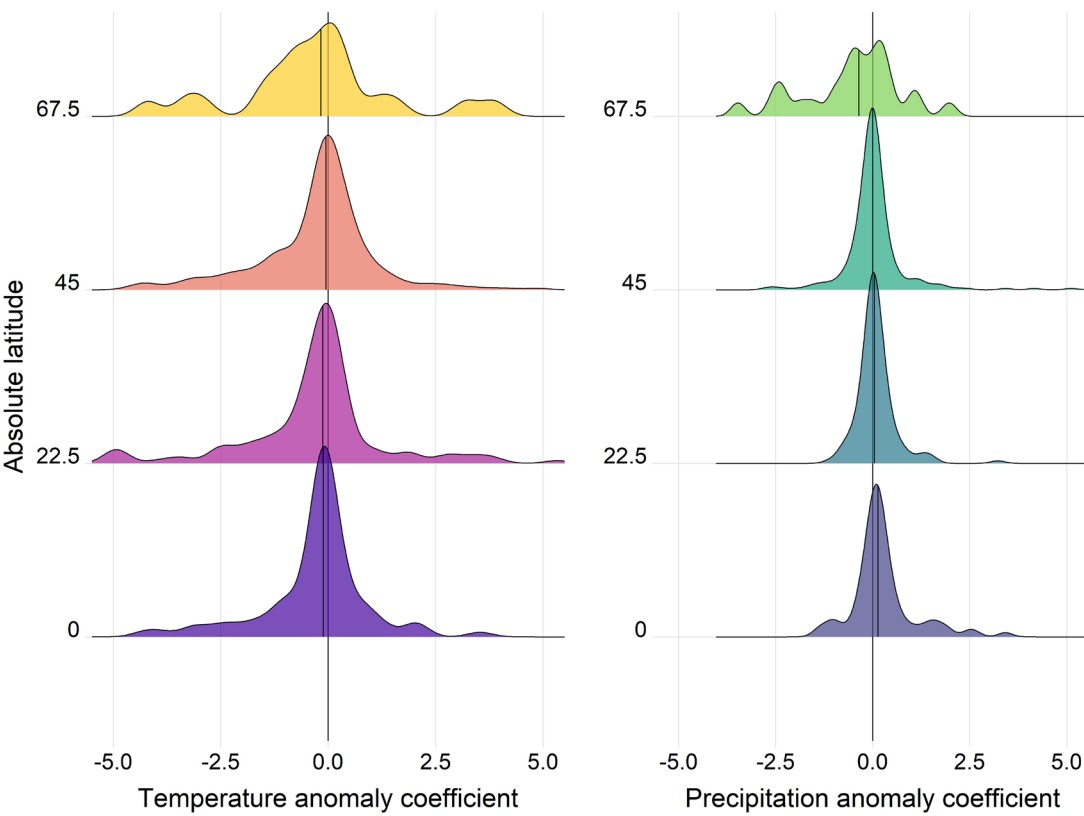

**Appendix 1—figure 17.** Density distributions of temperature (left) and precipitation (right) coefficients on abundance change with respect to the absolute latitude of the record location in the terrestrial mammals. Records were grouped based on their absolute latitude in categories of 22.5° e.g. 0 category indicates records found at absolute latitudes of 0-22.5°.

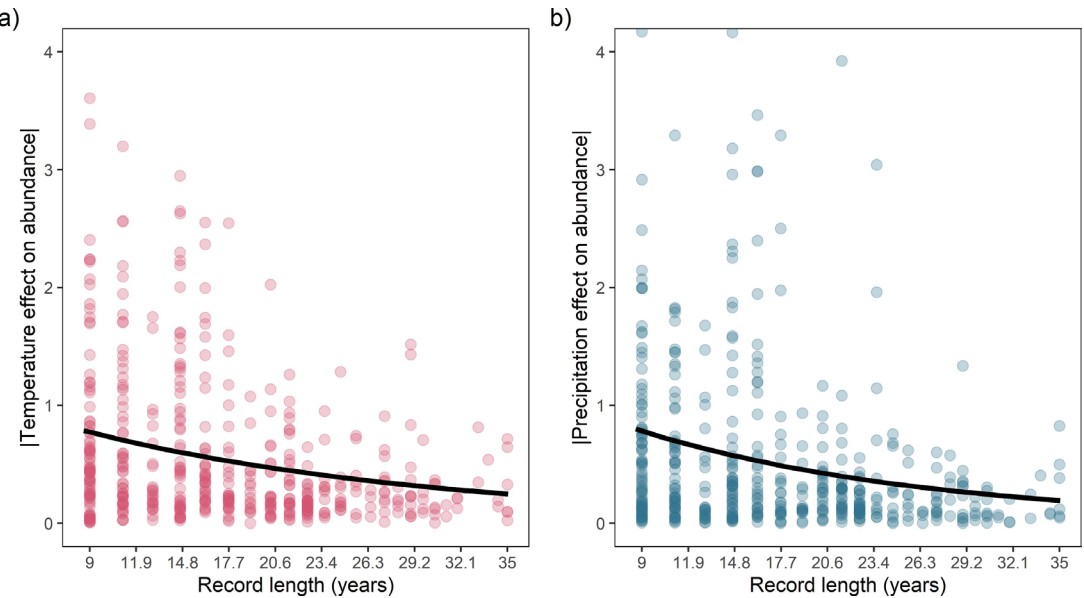

**Appendix 1—figure 18.** Posterior predictions for the influence of the record length on absolute temperature (**a**) and precipitation (**b**) effects on abundance changes in the terrestrial mammals. Points give the absolute weather effect for each record (N = 486). Only absolute weather effects <4 are displayed on the figure. Black lines are posterior means from the best predictive Gamma model including life-history effects.

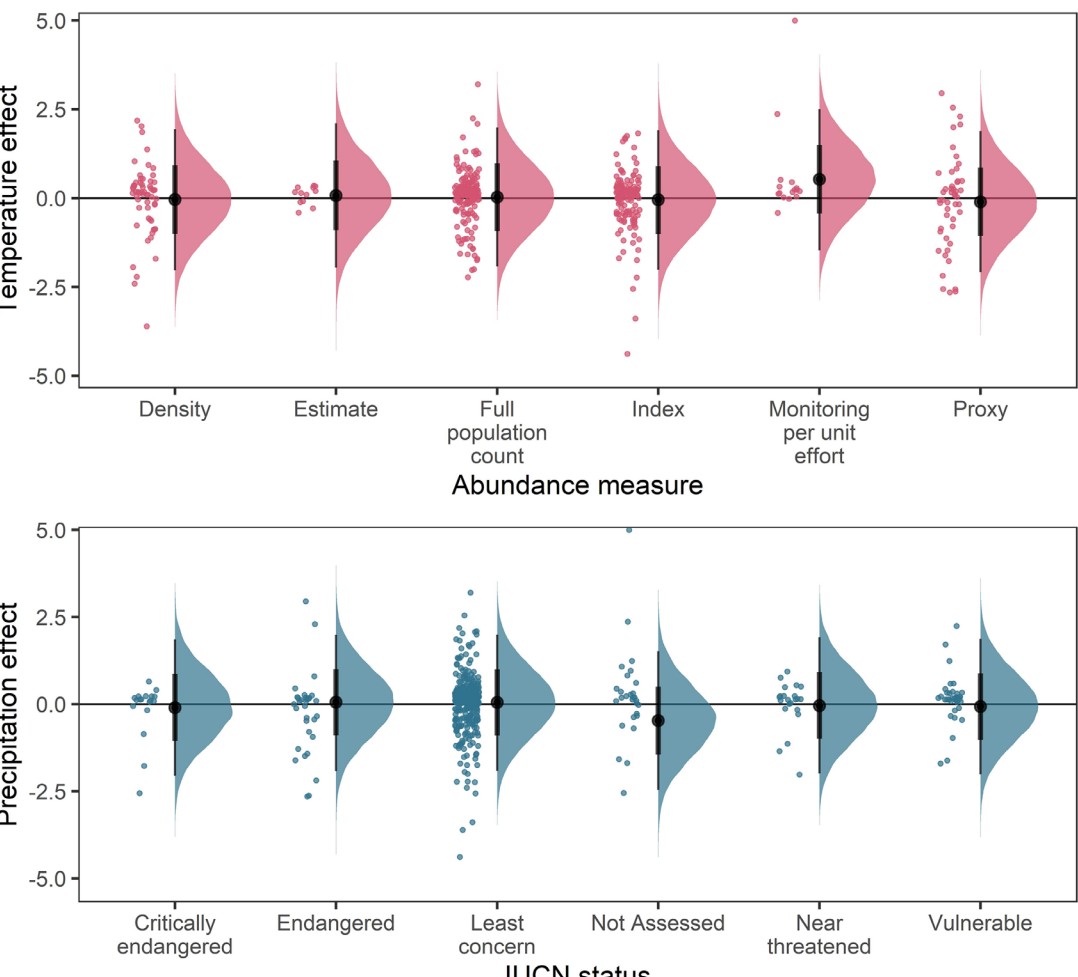

**Appendix 1—figure 19.** Posterior predictions for a Gaussian meta-regression model of temperature coefficients with the method of estimating abundance (top) and temperature coefficients with IUCN red-list status. To explore the potential impact of varying data-sources and reliability of the abundance data, we performed a further model selection to explore how the abundance measure type and IUCN red-list status influence temperature and precipitation responses (as in *equation 3*). For temperature coefficients the model including abundance measure type had a higher predictive performance than the base model (Δelpd = 3.13). For precipitation coefficients the model including IUCN red-list status had a higher predictive performance than the base model (Δelpd = 0.13). However, posterior predictions revealed that these predictive differences were not substantive, with posterior distributions centred on 0 for most abundance measure types (top) and IUCN statuses (bottom). The predictive differences are most likely a result of coefficients observed in the Monitoring per unit effort measure type (top) and the Not Assessed status (bottom), both of which have low sample size in the terrestrial mammals.

# Nearest neighbours plot

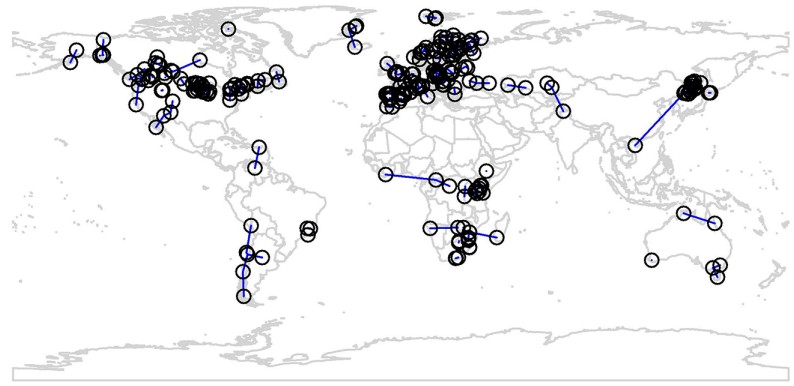

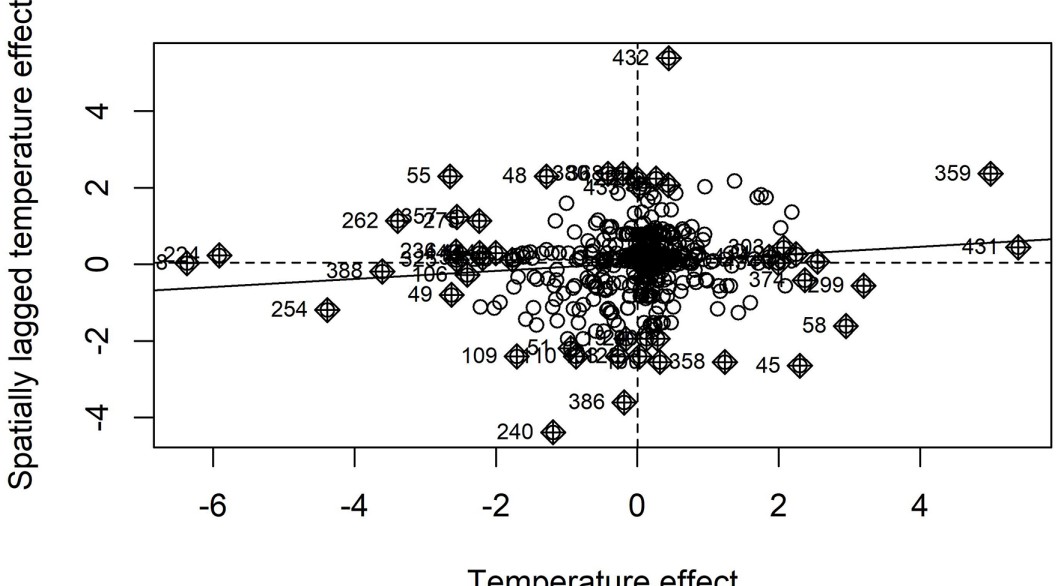

**Appendix 1—figure 20.** Nearest neighbour plot for abundance record localities and Moran's I plot for temperature coefficients. We explored spatial autocorrelation in the coefficients of the GAM models (*equation 2*) using Moran's I calculations with a nearest neighbours approach. Top – nearest neighbours plot for each abundance record in the study. We found low magnitude Moran's I for both temperature (I = 0.11, p = 0.03) and precipitation (I = 0.05, p > 0.05), but a significant Moran's I for temperature. Bottom – Moran's I plot for temperature effects indicates weak correlation between temperature effects and spatially lagged temperature effects, but the Moran's I plot indicates this is due to a small number of studies with high spatially lagged values.

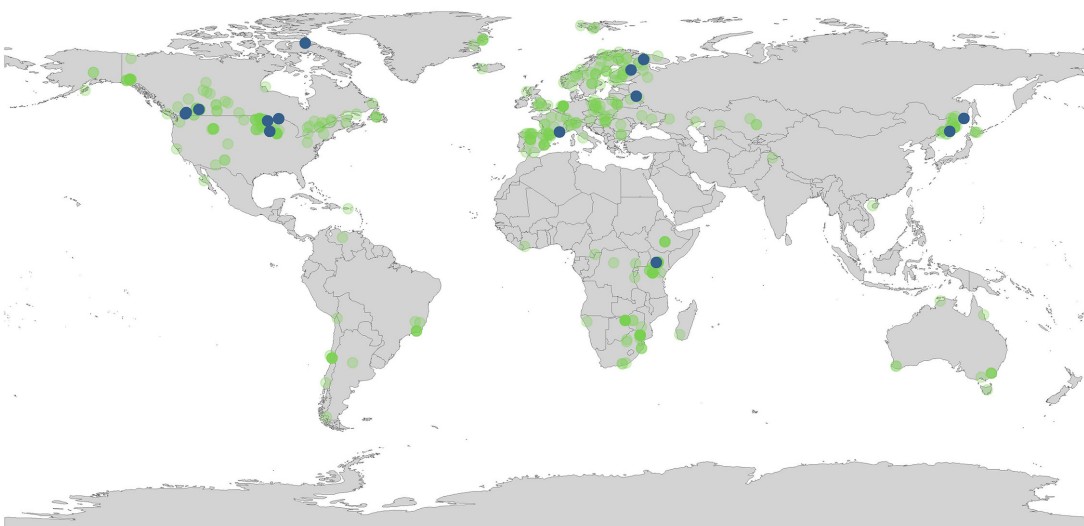

**Appendix 1—figure 21.** Local Moran's I significance for temperature coefficients. In addition to the global Moran's I analysis for temperature, we ran a local Moran's I for the temperature coefficients. Each point on the map gives the local Moran's I significance rating (95% level) and the spatial location of all records in the study. The local Moran's I indicates that a small number of spatially autocorrelated points is dictating general spatial autocorrelation patterns in the abundance records.

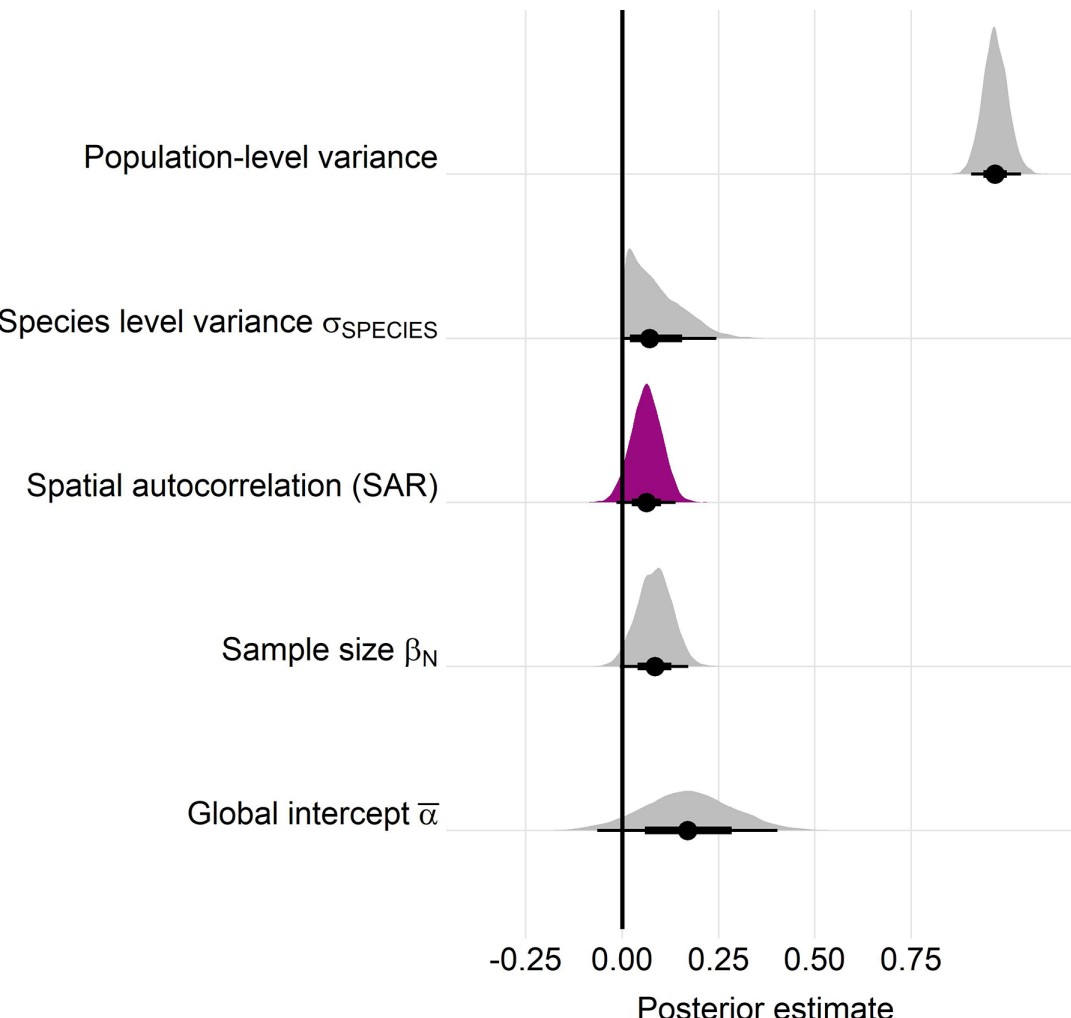

**Appendix 1—figure 22.** Posterior estimates for a Gaussian meta-regression model of temperature coefficients with an explicit spatially autocorrelated term across the terrestrial mammals. Posterior density distributions for each key model parameter (as in *equation 2*) given with half eye plots, where the point indicates the posterior average, and the bar is calculated using a cumulative distribution function. Purple density indicates the posterior estimate for the spatially autocorrelated term, whose posterior distribution overlapped with zero. Furthermore, leave-one-out cross validation indicated that the base model excluding spatial autocorrelation had a higher predictive performance than the model including the spatial term.

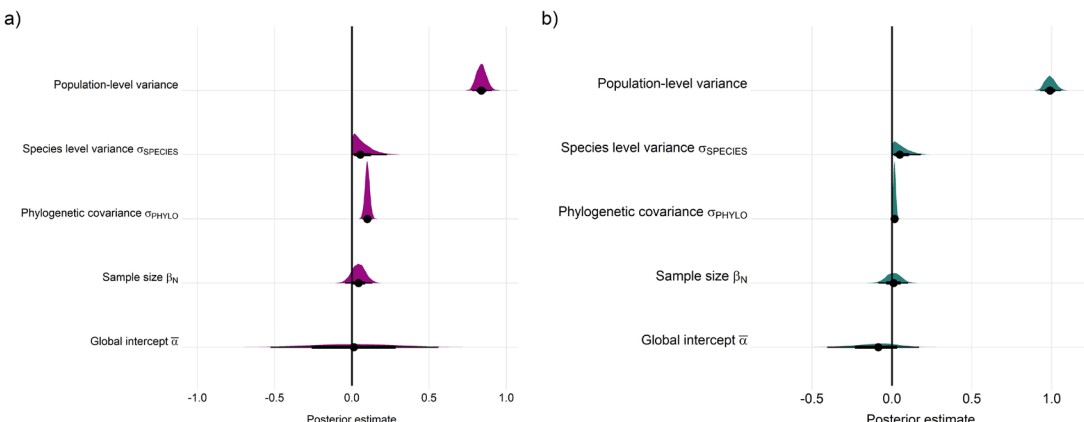

**Appendix 1—figure 23.** Posterior estimates for Gaussian meta-regression models for temperature (**a**) and precipitation (**b**) variance across the terrestrial mammals. Posterior density distributions for each key model parameter (as in *equation 2*) given with half eye plots, where the point indicates the posterior average, and the bar is calculated using a cumulative distribution function.

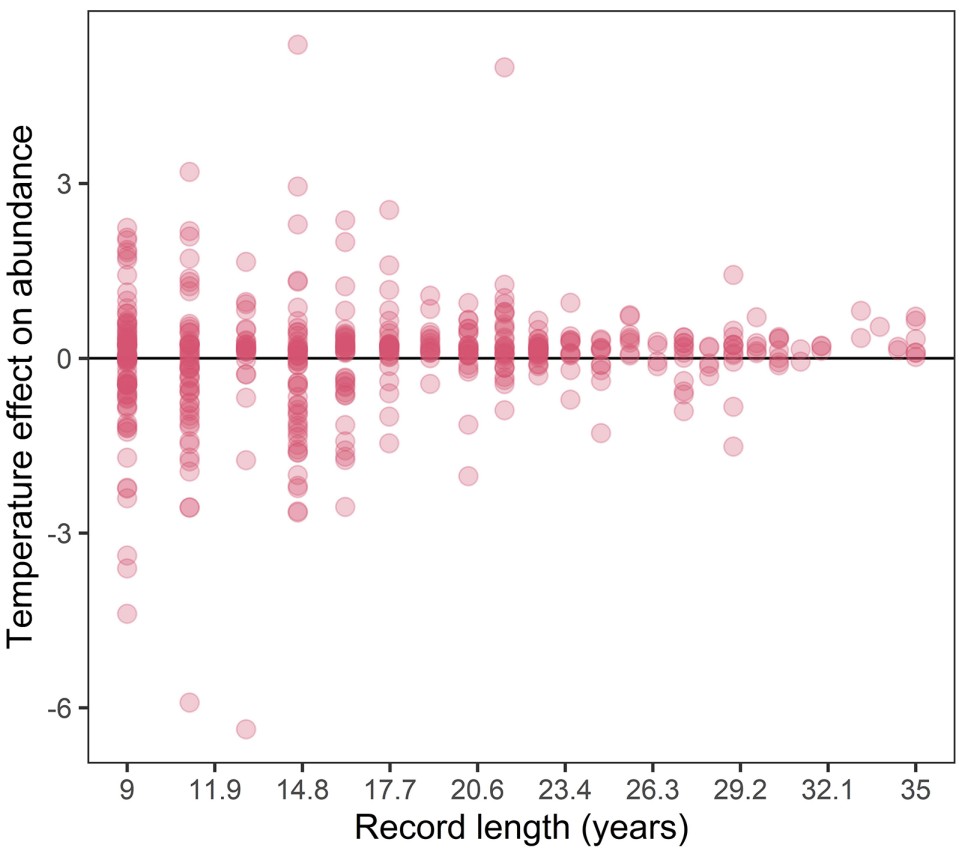

**Appendix 1—figure 24.** The association between record length and population responses to temperature anomalies in the terrestrial mammals. Points are the temperature responses of individual records. This figure highlights increased variance in temperature responses for shorter population records.

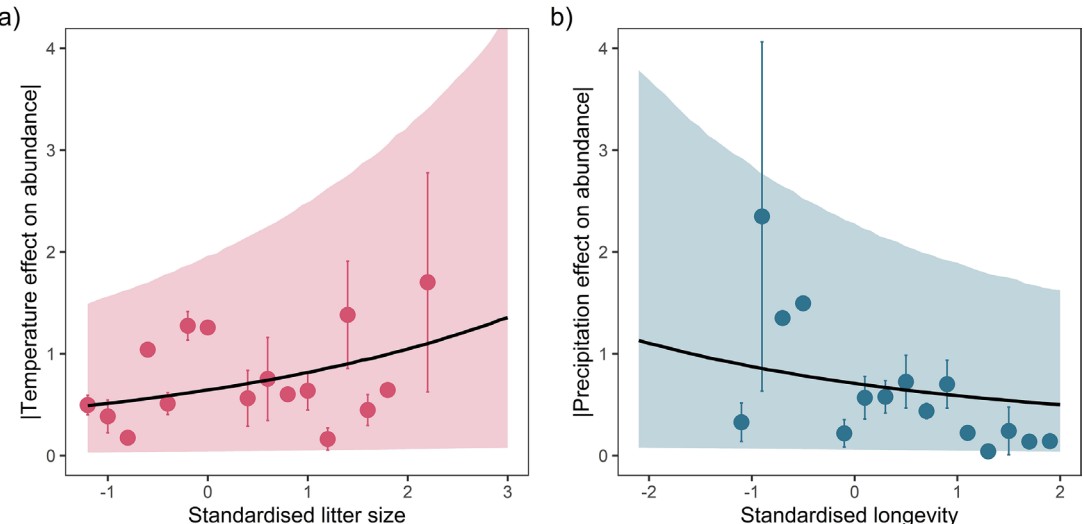

**Appendix 1—figure 25.** Posterior predictions of Gamma regressions between litter size and absolute responses to temperature (**a**), and maximum longevity and absolute responses to precipitation (**b**) for a high-quality subset of data ≥20 years. Standardisation was performed using z-scores of the natural-log of raw life-history traits. The values on each x-axis are split into equal bins of 0.2 units from the minimum to the maximum life-history value. Points are coefficient means, with standard error bars. The black lines are the mean posterior predictions from the best predictive model, where predictions were calculated averaging over all other covariates and varying effects in the model. The shaded intervals are the 90% quantile prediction intervals.

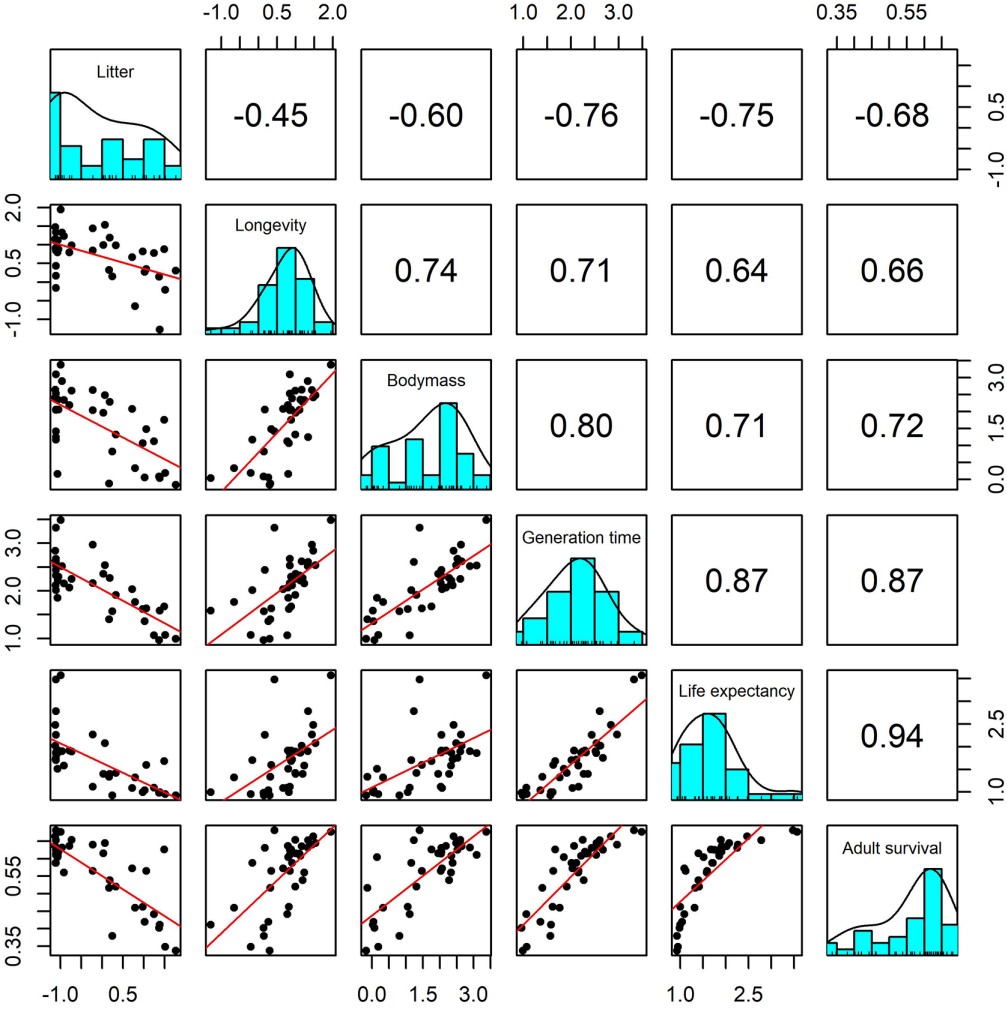

**Appendix 1—figure 26.** Pairwise Pearson correlation plots for all life-history traits explored for a subset of 37 terrestrial mammal species. Standardisation was performed using z-scores of the natural-log of raw life-history traits. Longevity, litter size and body mass we obtained from the sources highlighted in the Methods. The life-history traits generation time, life-expectancy and adult survival were obtained from the COMADRE database of animal matrix population models (Salguero-Gómez et al., 2016). Numbers are the Pearson's correlation coefficient. Generally, there was high covariance in all life-history traits.

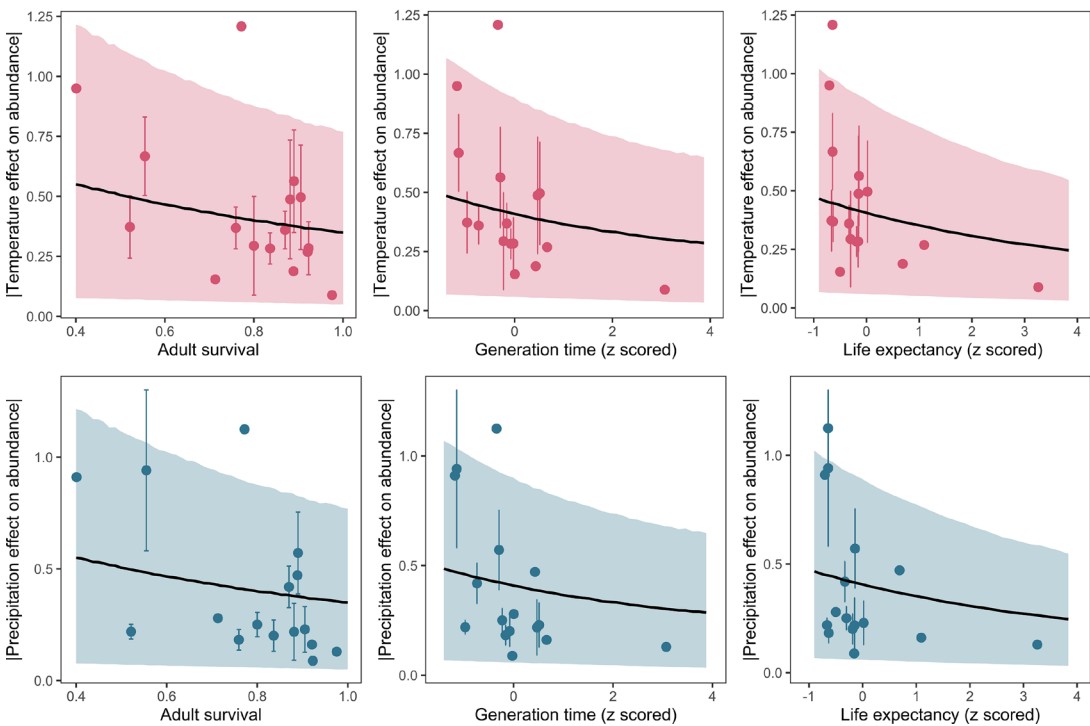

**Appendix 1—figure 27.** Posterior predictions of Gamma regressions between high-quality life-history traits from structured population models, and both temperature (top) and precipitation (bottom). Standardisation was performed using z-scores of the natural-log of raw life-history traits. The life-history traits were obtained from the COMADRE database of animal matrix population models (Salguero-Gómez et al., 2016). Points are coefficient means, with standard error bars. The black lines are the mean posterior predictions from the model, which excluded both phylogenetic and species-level variation. The shaded intervals are the 90% quantile prediction intervals.

