## [Editor Report]

Using 486 long-term population records of 157 mammal species, the authors show that species with a short life span and large litters are more affected, either positively or negatively, by extreme weather events than are species with a long life span and few offspring. This suggests that these "fast" species may require particular conservation attention, to avoid extinction due to the increased frequency and magnitude of extreme events.

---

## [Decision Letter]

**Decision letter after peer review:**

Thank you for submitting your article "Life-history predicts global population responses to the weather in the terrestrial mammals" for consideration by *eLife*.

Your article has been reviewed by three peer reviewers, and the evaluation has been overseen by Bernhard Schmid as the Reviewing Editor and Christian Rutz as the Senior Editor. The following individual involved in the review of your submission has agreed to reveal their identity: Nigel Yoccoz (Reviewer #1).

The reviewers have discussed their reviews with one another, and the Reviewing Editor has drafted this decision letter to help you prepare a revised submission.

Essential revisions:

1) Please try to improve the narrative by making it less exploratory (Reviewer #4). Clearly state your a priori hypotheses or at least explain why you expected the chosen life-history traits and environmental variables to be the most relevant and how they may interact and influence population dynamics. It may also be useful to explain why you expected robust prediction of absolute changes, but not for the direction of changes, and why this can still be useful for conservation biologists (Reviewer #4).

2) Please explain to which extent the greater absolute response of species to weather anomalies scales to a potentially larger absolute variation in population size of those species (Reviewer #2).

3) Please justify the selection of explanatory weather variables (Reviewers #3 and #4), e.g., by mentioning that fits with alternative variables were worse.

4) Perhaps try to analyze some subsets of the data to verify the robustness of the overall analysis (Reviewer #2).

5) Please discuss the limitations highlighted below by Reviewers #2-4, as well as the following points: traits were assumed constant between populations within species as were other characteristics of populations within species (e.g., marginality with regard to distribution range and thus "weather niche" (Reviewer #2), other traits (Reviewer #3) such as size at birth, sociality). Please provide a reason for why (or add corresponding analyses) biome (Reviewers #2 and #3) or site characteristics were not considered as explanatory variables (but rather subsumed in multilevel factors).

6) Please amend the title to read "Life-history predicts global population responses to weather in terrestrial mammals".

*Reviewer #1 (Recommendations for the authors):*

I must admit being a bit skeptical when it comes to large comparative analyses, as databases such as the Living Planet includes data of variable quality, and maximum longevity is, as you acknowledge, a far from ideal index for adult survival rate. Moreover, as you rightly point out, the available data are heavily biased towards some groups and biomes. Many analyses have been published using similar databases (e.g., Ono et al., 2019), so clearly some see value in such analyses. My concern is that because one has to "sell" the paper one avoids pointing out the limitations -- but alas this is not specific to your study.

One concern I have is how such analyses can be informed by the more detailed and accurate studies based on long-term monitoring of single populations (e.g., Gaillard et al., 2013). One knows that weather effects will vary among populations -- e.g., depending on where the population is in the climatic niche (see above) -- and such variation is crucial in predicting effects of weather and climate change. Large "meta-regressions" such as yours might be useful in summarizing the data available in one database, and you have done an excellent job analyzing these data, but given the far from random sample you have available, it is hard to assess how to make sense of the patterns observed. Perhaps using subsamples of high quality studies in different groups and biomes (if that is at all possible) to assess the predictive performance of your models may help.

You rightly discuss the limits of using maximum longevity as an index of a better measure such as adult survival rate. Would it be possible for the sample of species you have to at least assess the relationship between maximum longevity and adult survival rate? And for those species to repeat the analyses? The precision would be much lower as the sample size is likely to be small but you can at least assess the consistency of the relationships.

Regarding the effects of biomes, I did not understand how it was parameterized (e.g., l 465). It was included as a categorical predictor, right (figure S16 seems to indicate that)? So you have β_Biome[j] for j=1:number of biomes?

Adler, P. B., et al., 2011. Productivity Is a Poor Predictor of Plant Species Richness. Science 333:1750-1753.

Angert, A. L. 2009. The niche, limits to species' distributions, and spatiotemporal variation in demography across the elevation ranges of two monkeyflowers. Proceedings of the National Academy of Sciences of the United States of America 106 Suppl 2:19693-19698.

Gaillard, J. M., A. J. M. Hewison, F. Klein, F. Plard, M. Douhard, R. Davison, and C. Bonenfant. 2013. How does climate change influence demographic processes of widespread species? Lessons from the comparative analysis of contrasted populations of roe deer. Ecology Letters 16:48-57.

Ono, K., O. Langangen, and N. C. Stenseth. 2019. Improving risk assessments in conservation ecology. Nature Communications 10:7.

*Reviewer #2 (Recommendations for the authors):*

Jackson et al., present a global analysis of the effects of life history on the response of terrestrial mammal populations to weather, showing that litter size and longevity significantly alter how population's respond to anomalies in temperature and rainfall. The topic is highly interesting, and generally the manuscript is written in clear and concise way with some exceptions (see comments below). My main concern is about the timescales over which the weather events are calculated. As I understand it, these are anomalies away from the yearly expected value. This approach was of course used because the LPD data is recorded on an annual basis. However, there is a huge difference in the effect of slightly elevated temperatures across a whole year vs. one month with very high temperatures. The authors appear to have addressed this by looking at the yearly anomaly and the variance in the weather (annual weather variance) calculated across each year; however they are currently lacking some detail on exactly what this entailed (it is currently only mentioned in passing).

The authors have done a good job pulling all these analyses together into an accessible piece, and have provided all of the code to repeat these analyses.

*Reviewer #3 (Recommendations for the authors):*

1/ Overall framing and questions/hypotheses:

– Please justify more clearly, somewhere in the introduction, why looking at the weather is interesting for understanding future effects of climate change (see lines 87/88 and 111/112).

– In the introduction, the evidence from past work could be summarised more efficiently and concisely around key points (e.g., see lines 74-82, lines 96-104); thus the overall narrative/structure of the introduction can be improved.

– Beyond the three questions you ask, please explain what the underlying hypotheses are (e.g., for each question, adding a few words to explain what you expect: what are the spatial patterns you predict, what do you hypothesize would be the effect of life history, etc.)? In addition, the phrasing of the first question is a bit ambiguous with the term "consistent" (in my understanding, here you ask, across species and populations, whether the weather effects are significant, as in whether they differ from zero). Adding a specific hypothesis would also help to make this question clearer.

– Finally, please justify more why you look at the magnitude (absolute values) of the weather effects when investigating the influence of life-history on the responses, and why you don't investigate the directionality of the effects.

2/ Results

The results could be presented more efficiently and clearly overall. I would suggest including a few higher-level summaries throughout the manuscript to conclude on specific points.

3/ Discussion:

– Some of the limitations of the study should be developed in the discussion, in particular:

– The potential biases of the study (e.g., length of the time series): you mention that record length was significantly associated with temperature and precipitation effects. I would be good to mention whether your results are likely robust to these biases or otherwise what the expected effects on the results would be.

– The fact that you use absolute values (magnitude of the weather effects) for the life-history models, so that you can't conclude about the directionality of the effects. I suggest highlighting how your work helps predict future responses and inform conservation despite the fact that you didn't investigate the directionality of the effects (see lines 239/240; 243/244) (e.g., do your results imply that populations of species with fast life history should be more closely monitored irrespectively of whether they grow or decline?).

– It would be good for the discussion to make the key points stand out more.

4/ Methods

Life-history traits: It is important to include a statement about the degree of multicollinearity among the traits.

---

## [Author Response]

Essential revisions:1) Please try to improve the narrative by making it less exploratory (Reviewer #4). Clearly state your a priori hypotheses or at least explain why you expected the chosen life-history traits and environmental variables to be the most relevant and how they may interact and influence population dynamics. It may also be useful to explain why you expected robust prediction of absolute changes, but not for the direction of changes, and why this can still be useful for conservation biologists (Reviewer #4).

We have addressed this point by thoroughly revising the introduction to include a specific set of hypotheses relating to both overall weather responses and interactions with species life-history L133-144. We also include a justification for why we included absolute changes L138-141, and why this is still important for conservation L258-262.

2) Please explain to which extent the greater absolute response of species to weather anomalies scales to a potentially larger absolute variation in population size of those species (Reviewer #2).

We have addressed this point with an extensive response to reviewer #1 (here reviewer #2). We feel that given our use of state-space models that explicitly model process noise (variation in population size), we have largely accounted for this variation. We now state this in the methods L531-535. However, we also argue that this variation is a core component of what we are trying to capture i.e. we expect that short-lived species have greater variability in population size as a result of their evolved response to the environment. We hope that with the more explicit hypotheses included here this is made clearer.

3) Please justify the selection of explanatory weather variables (Reviewers #3 and #4), e.g., by mentioning that fits with alternative variables were worse.

We have addressed this point by more explicitly highlighting our additional analyses on weather variance as well as weather anomalies. We have now included a clearer narrative of weather variance in the introduction L90-94, as well as explicit indication of these results L173-174. Our exploration of weather variance yielded near-identical results to that of the weather anomalies, and so we performed later analyses with weather anomalies.

4) Perhaps try to analyze some subsets of the data to verify the robustness of the overall analysis (Reviewer #2).

We have addressed this point by providing extensive additional analysis to explore the implications of our selection of life-history traits (by using more detailed life-history traits from structured population models) and our use of long-term records (by exploring both higher quality and lower quality subsets). These additional analyses are detailed in the Methods L401-402, L454-465 and results are included as supplementary figures Figure S25-27.

5) Please discuss the limitations highlighted below by Reviewers #2-4, as well as the following points: traits were assumed constant between populations within species as were other characteristics of populations within species (e.g., marginality with regard to distribution range and thus "weather niche" (Reviewer #2), other traits (Reviewer #3) such as size at birth, sociality). Please provide a reason for why (or add corresponding analyses) biome (Reviewers #2 and #3) or site characteristics were not considered as explanatory variables (but rather subsumed in multilevel factors).

We have addressed this comment primarily by including an explicit limitation section of the discussion L350-366 in which we describe the key limitations of the current study in detail. We hope that this in conjunction with our conservative and extensive modelling framework are convincing to address these limitations (although not fully, as pointed out by all reviewers). On the point about weather niche, we have now included additional discussion on within-species variation L330-340. Finally, we have amended our lack of clarity by specifying that biomes were explicitly included as explanatory variables L185-190.

6) Please amend the title to read "Life-history predicts global population responses to weather in terrestrial mammals".

This point has been addressed.

Reviewer #1 (Recommendations for the authors):I must admit being a bit skeptical when it comes to large comparative analyses, as databases such as the Living Planet includes data of variable quality, and maximum longevity is, as you acknowledge, a far from ideal index for adult survival rate. Moreover, as you rightly point out, the available data are heavily biased towards some groups and biomes. Many analyses have been published using similar databases (e.g., Ono et al., 2019), so clearly some see value in such analyses. My concern is that because one has to "sell" the paper one avoids pointing out the limitations -- but alas this is not specific to your study.

We thank the reviewer for their honesty and scepticism for large comparative analyses. In general, we do agree that data quality and biases mean that rigor and conservatism in comparative analyses is crucial. Throughout the progression of this study, we have used conservative Bayesian inference with regularising priors, performed supplementary and confirmatory analyses, and ensured that we performed extensive data checks (please see supplementary code). However, we agree that acknowledging the key limitations of the study is crucial and was lacking in the previous version of the manuscript. In response to Reviewer #3 below, we have now included a dedicated limitations section in the discussion L350-366.

One concern I have is how such analyses can be informed by the more detailed and accurate studies based on long-term monitoring of single populations (e.g., Gaillard et al., 2013). One knows that weather effects will vary among populations -- e.g., depending on where the population is in the climatic niche (see above) -- and such variation is crucial in predicting effects of weather and climate change. Large "meta-regressions" such as yours might be useful in summarizing the data available in one database, and you have done an excellent job analyzing these data, but given the far from random sample you have available, it is hard to assess how to make sense of the patterns observed. Perhaps using subsamples of high quality studies in different groups and biomes (if that is at all possible) to assess the predictive performance of your models may help.

Reviewer #1 and Reviewer #3 both raise an excellent point on testing our findings for high quality subsets of the data. In particular, temporal replication is a component that we believe will give more indications of the robustness of our analyses. We have therefore run extensive additional analyses to explore the implications of data quality on our findings. Specifically, we repeated the full meta-regression framework, but included ≥5 years or ≥20 years of abundance data. The latter in particular enabled us to critically evaluate the assumptions of our key findings with respect to data quality. In both cases, in conjunction with our main findings we did not find evidence for directional patterns of weather responses. Posterior estimates temperature and precipitation anomalies are given in Author response image 1.

**Author response image 1. sa2fig1:** 

Interestingly, for both temperature and precipitation, models with ≥5 years of abundance data had greater estimates of phylogenetic covariance with respect to species-level variance. This is probably a result of reduced repeated observations.We then repeated life-history model selection for both data subsets. For temperature, we found that including life-history variables did not improve model predictive performance for records ≥5 years in length. Model selection results can be found in Author response table 1.

**Author response table 1. sa2table1:** 

Model	predictors	LOO elpd	LOO elpd error	elpd difference	elpd error difference	LOO information criterion
temp_base_5yr	base	840.15	46.76	0.00	0.00	-1,680.31
temp_lh_uni_5yr	longevity+ bodymass + litter	839.90	47.11	-0.25	1.34	-1,679.81

For precipitation, there was a slight increase in predictive performance for the model including life-history variables, however this was only a marginal change and both longevity and litter coefficients were moderate and substantially overlapped 0.

**Author response table 2. sa2table2:** 

Model	predictors	LOO elpd	LOO elpd error	elpd difference	elpd error difference	LOO information criterion
precip_lh_uni_5yr	longevity+ bodymass + litter	796.61	42.92	0.00	0.00	-1,593.21
precip_base_5yr	base	795.98	42.95	-0.62	0.96	-1,591.96

In summary, we found that while key directional patterns remained in the data subset with decreased numbers of observations, this was not reflected in life-history patterns, for which we conclude a more detailed subset of data was necessary to capture information on weather responses.

Repeated the analyses for the high-quality subset of data ≥20 years, overall the negative impact of longevity on absolute weather responses and positive impact of litter size was maintained. In this analysis, given a smaller subset of 54 species and 94 observations, we concluded that it was more pragmatic to ignore phylogenetic covariance and explore only within-species variance. In this repeated analysis, for both temperature and precipitation we found predictive support for life-history traits relative to the base model.

**Author response table 3. sa2table3:** 

Model	predictors	LOO elpd	LOO elpd error	elpd difference	elpd error difference	LOO information criterion
temp_litter_20yr_nophylo	litter	-40.95	10.18	0.00	0.00	81.89
temp_lh_uni_20yr_nophylo	longevity +bodymass + litter	-42.06	10.25	-1.11	0.32	84.12
temp_longevity-20yr-nophylo	longevity	-42.99	10.64	-2.05	1.98	85.99
temp_base_20yr_nophylo	base	-43.85	11.14	-2.91	2.25	87.71

**Author response table 4. sa2table4:** 

Model	predictors	LOO elpd	LOO elpd error	elpd difference	elpd error difference	LOO information criterion
precip_longevity_20yr_nophylo	longevity	-22.49	11.53	0.00	0.00	44.97
precip_base_20yr_nophylo	base	-22.95	12.35	-0.46	1.70	45.90
precip_litter_20yr_nophylo	litter	-23.09	11.64	-0.60	0.90	46.18
precip_lh-uni-20yr_nophylo	longevity _+ bodymass + litter_	-24.06	11.79	-1.57	1.23	48.12

However, life-history coefficients were not as pronounced in this smaller subset. For temperature, there was still a strong positive association with litter size, but the longevity effect was reduced. In contrast, for precipitation the negative longevity effect remained but the litter size effect diminished. Posterior predictions of these patterns can be found in Author response image 2.

Overall, these additional analyses with both increased and decreased data quality revealed consistent patterns with our main findings. Importantly however, we suggest that ≥5 years of data are not sufficient to characterise annual responses to weather.In answer to the original point of the reviewer, rerunning analyses on a subset of increased quality (longer records) yielded consistent results in terms of life-history. Namely, traits relating to the fast-slow continuum were still associated with absolute responses to weather anomalies. We therefore conclude that our broad findings are robust. We have now included a brief description of this additional analysis L401-403 and the inclusion of a supplementary Figure S25.

You rightly discuss the limits of using maximum longevity as an index of a better measure such as adult survival rate. Would it be possible for the sample of species you have to at least assess the relationship between maximum longevity and adult survival rate? And for those species to repeat the analyses? The precision would be much lower as the sample size is likely to be small but you can at least assess the consistency of the relationships.

The reviewer rightly points out the issues with maximum longevity, which of course as a measure of the lifespan of the oldest recorded individual of a species, introduces a lot of noise when assessing species-level life history. We completely agree that in an ideal world we would have a fully formed life-table/structured population model with which we could assess other key life-history traits such as adult survival, life expectancy or generation time (Gaillard et al., 2005). The question is then whether the noise introduced by this biased measure is too much, given the research questions. Our goals were comparative here, which the reviewer concedes they have scepticism in, but that we feel are important to explore broad global patterns with available data. The goal of this study was to explore weather responses for as many mammal species as possible, for which we felt we needed life-history variables that gave us a comparable breadth of data. We reasoned that the broad metrics of maximum longevity and mean litter size were largely characteristic of key life-history axes for this number of species. However, as is rightly pointed out we did not initially explore this assumption explicitly.

Therefore, we have now performed several additional analyses to address these concerns, which can now be found in methods section L454-465 and with additional supplementary figures Figure S26 and Figure S27. We did not find that the choice of broad life-history metrics impacted our key results.

To answer in more detail here however, as is pointed out by the reviewer, this level of demographic detail is hard to come by, save for high-quality, individual-based, longitudinal studies, which the reviewer is extremely familiar with. We opted to use the COMADRE animal matrix population model database to extract more detailed demographic information for as many terrestrial mammal species as possible. After data checks (including only ergodic, reducible, primitive, non-NA matrices etc.) we were left with 37 species that had both detailed structured population models and the broad metrics of life-history we include. From these models, we calculated generation time (the mean number of years needed for a cohort to replace itself, a key fast-slow trait used in many studies, see Healy *et al.,* (2019) for a recent example), life expectancy, and adult survival (here defined as the mean survival of adult stages in the matrix). First, we explored the collinearity between these life-history variables in the larger subset of 37 species, which is presented in Figure S26. This pairs plot gives the Pearson’s correlation coefficients for each pairwise set of log-transformed life history variables that we assessed here.

You will see that, in spite of the biases highlighted, the life-history variables included here are all highly correlated with one another, even for this small subset of 37 species. Specifically for Maximum Longevity, it has Pearson’s coefficients of >0.64 when correlated with more detailed life-history variables. This suggests that our choice of broad characteristics is appropriate for this comparative question.

As suggested, we then extended to include these life-history variables in our meta-regression framework. However, the number of species that had both sufficient life-history information and LPD population data and weather responses was 16 (10 times lower than the number of species included in the main findings of the manuscript). We hope you agree that this amount of data is not ideal to explore these patterns at a global scale. Therefore, we analysed this data in a much simpler hierarchical model framework rather than the full phylogenetically controlled analyses. Here, we simply regressed life history variables against absolute weather responses, including only a random effect of species and controlling for sample size. You can see the results of the posterior predictions in Figure S26. Posterior predictions are given with the 90% credible intervals for each life-history/weather effect combination.

While the sample size here prohibits a more detailed exploration of these patterns, the same negative relationships can be observed. The coefficients of these responses were all below -0.12. These reiterate our findings of increased absolute responses to weather in short-lived species.

Gaillard, J.M., Yoccoz, N.G., Lebreton, J.D., Bonenfant, C., Devillard, S., Loison, A. et al.,

(2005). Generation time: a reliable metric to measure life-history variation among mammalian populations. Am. Nat., 166, 119–123

Healy, K., Ezard, T.H.G., Jones, O.R. et al., Animal life history is shaped by the pace of life and the distribution of age-specific mortality and reproduction. Nat Ecol Evol 3, 1217–1224 (2019). https://doi.org/10.1038/s41559-019-0938-7

Regarding the effects of biomes, I did not understand how it was parameterized (e.g., l 465). It was included as a categorical predictor, right (figure S16 seems to indicate that)? So you have β_Biome[j] for j=1:number of biomes?

The reviewer Is correct, biome was included as a categorical predictor. This point has been addressed L189, with further explanations throughout the text explaining our use of biomes (following comments from other reviewers). Technically, the reference to the sub-script *j* is incorrect here, as this is in reference to a species (L555). However, the equation form described is correct and so we have added this additional information to equation 3 L564-567.

Adler, P. B., et al., 2011. Productivity Is a Poor Predictor of Plant Species Richness. Science 333:1750-1753.Angert, A. L. 2009. The niche, limits to species' distributions, and spatiotemporal variation in demography across the elevation ranges of two monkeyflowers. Proceedings of the National Academy of Sciences of the United States of America 106 Suppl 2:19693-19698.Gaillard, J. M., A. J. M. Hewison, F. Klein, F. Plard, M. Douhard, R. Davison, and C. Bonenfant. 2013. How does climate change influence demographic processes of widespread species? Lessons from the comparative analysis of contrasted populations of roe deer. Ecology Letters 16:48-57.Ono, K., O. Langangen, and N. C. Stenseth. 2019. Improving risk assessments in conservation ecology. Nature Communications 10:7.Reviewer #2 (Recommendations for the authors):Jackson et al., present a global analysis of the effects of life history on the response of terrestrial mammal populations to weather, showing that litter size and longevity significantly alter how population's respond to anomalies in temperature and rainfall. The topic is highly interesting, and generally the manuscript is written in clear and concise way with some exceptions (see comments below). My main concern is about the timescales over which the weather events are calculated. As I understand it, these are anomalies away from the yearly expected value. This approach was of course used because the LPD data is recorded on an annual basis. However, there is a huge difference in the effect of slightly elevated temperatures across a whole year vs. one month with very high temperatures. The authors appear to have addressed this by looking at the yearly anomaly and the variance in the weather (annual weather variance) calculated across each year; however they are currently lacking some detail on exactly what this entailed (it is currently only mentioned in passing).

We apologise for the lack of detail when describing the effect of weather variance. The reviewer is absolutely correct, we explored the effect of weather variance on population growth in addition to weather anomalies. Generally, we aimed to explore the consequences of ‘extreme’ weather in the current study, hence the exploration of anomalies and variance in contrast to climate trends for example. As pointed out by reviewer #1, this focus on weather was not entirely clear in the introduction, and we have therefore more explicitly made the link to weather anomalies and weather variance in the new version L90-94. In Figure S23 we presented the posterior estimates for the effect of temperature (a) and precipitation (b) variance on population growth rates.

These results are near identical to the results for weather anomalies, with the same conclusion of no directional pattern of temperature or precipitation variance on population growth rates. The model for temperature does highlight an interesting pattern of increase phylogenetic covariance however.

In the updated manuscript, we have made this result clearer L173-174. Furthermore, we now explicitly refer to weather variance in the introduction, including a brief description, when outlining our methodology L120-122.

Other than this I have few significant comments -- the authors have done a good job pulling all these analyses together into an accessible piece, and have provided all of the code to repeat these analyses.

We thank the reviewer for their positive feedback, which is very much appreciated and rewarding given many of our goals on the paper.

Reviewer #3 (Recommendations for the authors):1/ Overall framing and questions/hypotheses:– Please justify more clearly, somewhere in the introduction, why looking at the weather is interesting for understanding future effects of climate change (see lines 87/88 and 111/112).

This point has been addressed in the Introduction with more information on investigating weather as opposed to climate L80-97. For the full response to this point please see the response to reviewer #1 above.

– In the introduction, the evidence from past work could be summarised more efficiently and concisely around key points (e.g., see lines 74-82, lines 96-104); thus the overall narrative/structure of the introduction can be improved.

In addition to addressing other points in the introduction, we have thoroughly reworked the sections mentioned in this comment. We aimed to make these specific points much more concise and clearer in narrative, which you will now find L68-151. We hope you agree that these improve the manuscript. However, given the additional changes to add hypotheses, justification for weather, and more detail on the method intro (Reviewer #2), we have restructured large sections of the introduction.

– Beyond the three questions you ask, please explain what the underlying hypotheses are (e.g., for each question, adding a few words to explain what you expect: what are the spatial patterns you predict, what do you hypothesize would be the effect of life history, etc.)?

We have now extensively reworked this section of the introduction to clearly highlight our main four hypotheses, L133-144. Generally, we expected no directional effect (following the winner/loser paradigm introduced in the second paragraph), but greater absolute responses in ‘fast’ species (paragraph 4 – following Morris et al., 2008 and Compagnoni et al., 2021). Given the demographic components underpinning population change, which are evolutionarily constrained (James et al., 2020), we initially expected phylogenetic autocorrelation. Finally, although spatial effects are less clear, we predicted that biomes with lower climate variability, where organisms are evolved in narrow climatic niches, would have larger impacts of weather anomalies. Please see introduction for full response.

In addition, the phrasing of the first question is a bit ambiguous with the term "consistent" (in my understanding, here you ask, across species and populations, whether the weather effects are significant, as in whether they differ from zero). Adding a specific hypothesis would also help to make this question clearer.

On reflection for this point, we agree that the term ‘consistent’ was not necessarily indicative of what we were attempting to communicate. And the reviewer rightly points out that we were aiming to ascertain whether responses were different from 0. However, we do not agree that the term significant is appropriate in the context of the current study given its heavy Bayesian focus, which may result in confusion with inference. Instead, we have opted to use the word directional in the manuscript, and have clarified what this means explicitly in the introduction L125-126 and Methods L550-551.

– Finally, please justify more why you look at the magnitude (absolute values) of the weather effects when investigating the influence of life-history on the responses, and why you don't investigate the directionality of the effects.

This choice to explore only the absolute magnitude of responses with respect to life-history was two-fold: We do not think there is good *a-priori* reasoning for a particular life-history to have positive/negative responses specifically (Morris et al., 2008), and this has been the approach in the previously published work of Compagnoni et al., (2021), for which absolute population coefficients were also explored with respect to weather (see also Le Coeur et al., 2021 for which absolute values of the long-term population growth rate were investigated with respect to interannual weather variance). While this reasoning was included in the discussion in our initial submission L591-593, we agree that this information is useful to place in the Introduction of the manuscript. Therefore we have now included this justification L138-141.

Le Coeur, C., Storkey, J., and Ramula, S. (2021). Population responses to observed climate variability across multiple organismal groups. *Oikos*, *130*(3), 476-487.

2/ ResultsThe results could be presented more efficiently and clearly overall. I would suggest including a few higher-level summaries throughout the manuscript to conclude on specific points.

Following this suggestion, we have re-structured the Results section to clear highlight results at the opening of each paragraph, specifically at L165-166, L186-188, L211-212 and L232-234. We hope this presents the key findings more efficiently.

3/ Discussion:– Some of the limitations of the study should be developed in the discussion, in particular:

We agree that acknowledging the limitations in the current study (which as all reviewers point out, is common to macroecological studies) is very important. Therefore, we have now included an additional section in the discussion highlighting these limitations explicitly L350-366. Further clarifications are included in response below, but please see the revised Discussion section for full details.

– The potential biases of the study (e.g., length of the time series): you mention that record length was significantly associated with temperature and precipitation effects. I would be good to mention whether your results are likely robust to these biases or otherwise what the expected effects on the results would be.

The reviewer raises an important point regarding the length of the time series, which is certainly important to acknowledge. Given that record length was controlled for in all analyses, and we did not have a-priori reasoning for expecting it to be associated with life-history, we conclude that our results are not impacted by this. Please see results L174-183 for this. However, we also now include this explicitly in the limitations paragraph of the discussion L350-366.

– The fact that you use absolute values (magnitude of the weather effects) for the life-history models, so that you can't conclude about the directionality of the effects. I suggest highlighting how your work helps predict future responses and inform conservation despite the fact that you didn't investigate the directionality of the effects (see lines 239/240; 243/244) (e.g., do your results imply that populations of species with fast life history should be more closely monitored irrespectively of whether they grow or decline?).

The reviewer is absolutely correct, and we agree about the monitoring interpretation. We have added this specific point to the first paragraph of the discussion L258-262. Specifically regarding the result however, we are not able to conclude on the directionality of weather effects with respect to life-history. Please see response above for full justification of this. Following this point however, we have added this point to the limitation section of the discussion L350-366.

– It would be good for the discussion to make the key points stand out more.

We have now made every effort to improve the clarity of the discussion. To achieve this, we have (1) included a ‘summary’ first paragraph L251-262 that highlights the study’s core results and implications, based on the comment of the reviewer above, (2) restructured the second paragraph L263-273 to make the key points stand out more clearly, (3) added summary sentences highlighting key results L263 and L297-298, (4) included a limitations paragraph to clearly highlight the study’s short-comings.

4/ MethodsLife-history traits: It is important to include a statement about the degree of multicollinearity among the traits.

We agree that the collinearity of the life-history variables is a very important aspect of this study to explore and account for. Allometric relationships in particular are very pervasive in macroecology and evolution and so investigating how lifespan and litter size influence population size independently of body size is critical.

To address this point explicitly, while this information was included on the Zenodo repository plot/lifehistory_raw/, we now include a statement of the collinearity in our life-history variables i.e. Pearson’s regression coefficients with adult body mass L459-465. However, in addition to these changes, we made further changes on the request of Reviewer #1 to incorporate more detailed demographic information. Therefore this section of the methods was thoroughly revised. For our full response on the appropriate use of life-history variables, please see the response to Reviewer #1 above.